# Designed CXCR4 mimic acts as a soluble chemokine receptor that blocks atherogenic inflammation by agonist-specific targeting

Christos Kontos[1,11], Omar El Bounkari[2,11], Christine Krammer[2,11], Dzmitry Sinitski[2], Kathleen Hille[1], Chunfang Zan[2], Guangyao Yan[2], Sijia Wang[2], Ying Gao[2], Markus Brandhofer[2], Remco T. A. Megens[3], Adrian Hoffmann[2,4], Jessica Pauli [5], Yaw Asare [2], Simona Gerra[2], Priscila Bourilhon [2], Lin Leng[6], Hans-Henning Eckstein[5], Wolfgang E. Kempf[5], Jaroslav Pelisek[5,7], Ozgun Gokce[2,8], Lars Maegdefessel[5], Richard Bucala[6], Martin Dichgans [2,8], Christian Weber[3,8,9,10], Aphrodite Kapurniotu [1,12✉] & Jürgen Bernhagen [2,8,9,12✉]

Targeting a specific chemokine/receptor axis in atherosclerosis remains challenging. Soluble receptor-based strategies are not established for chemokine receptors due to their discontinuous architecture. Macrophage migration-inhibitory factor (MIF) is an atypical chemokine that promotes atherosclerosis through CXC-motif chemokine receptor-4 (CXCR4). However, CXCR4/CXCL12 interactions also mediate atheroprotection. Here, we show that constrained 31-residue-peptides ('msR4Ms') designed to mimic the CXCR4-binding site to MIF, selectively bind MIF with nanomolar affinity and block MIF/CXCR4 without affecting CXCL12/CXCR4. We identify msR4M-L1, which blocks MIF- but not CXCL12-elicited CXCR4 vascular cell activities. Its potency compares well with established MIF inhibitors, whereas msR4M-L1 does not interfere with cardioprotective MIF/CD74 signaling. In vivo-administered msR4M-L1 enriches in atherosclerotic plaques, blocks arterial leukocyte adhesion, and inhibits atherosclerosis and inflammation in hyperlipidemic $Apoe^{-/-}$ mice in vivo. Finally, msR4M-L1 binds to MIF in plaques from human carotid-endarterectomy specimens. Together, we establish an engineered GPCR-ectodomain-based mimicry principle that differentiates between disease-exacerbating and -protective pathways and chemokine-selectively interferes with atherosclerosis.

[1] Division of Peptide Biochemistry, TUM School of Life Sciences, Technische Universität München (TUM), 85354 Freising, Germany. [2] Institute for Stroke and Dementia Research (ISD), Klinikum der Universität München, Ludwig-Maximilians-Universität (LMU) München, 81377 Munich, Germany. [3] Institute for Cardiovascular Prevention, Klinikum der Universität München, Ludwig-Maximilians-Universität (LMU) München, 80336 Munich, Germany. [4] Department of Anaesthesiology, Klinikum der Universität München, Ludwig-Maximilians-Universität (LMU) München, 81377 Munich, Germany. [5] Department for Vascular and Endovascular Surgery, Klinikum rechts der Isar, Technische Universität München (TUM), 81675 Munich, Germany. [6] Yale University School of Medicine, New Haven, CT 06510, USA. [7] Department of Vascular Surgery, University Hospital Zurich, 8091 Zurich, Switzerland. [8] Munich Cluster for Systems Neurology (SyNergy), 81377 Munich, Germany. [9] Munich Heart Alliance, 80802 Munich, Germany. [10] Cardiovascular Research Institute Maastricht (CARIM), Maastricht University, 6229 Maastricht, The Netherlands. [11]These authors contributed equally: Christos Kontos, Omar El Bounkari, Christine Krammer. [13]These authors jointly supervised this work: Aphrodite Kapurniotu, Jürgen Bernhagen. ✉email: akapurniotu@wzw.tum.de; juergen.bernhagen@med.uni-muenchen.de

C hemokines are chemotactic cytokines that orchestrate cell trafficking and behavior in homeostasis and disease. Four chemokine sub-classes and their G protein-coupled receptor (GPCR)-type receptors (CKRs) constitute a complex ligand/receptor-network characterized by both specificity and redundancy[1,2]. Chemokines are pivotal players in various inflammatory diseases, including atherosclerosis[1,3]. Therapeutic anti-cytokine approaches are successfully used in several inflammatory diseases and the positive results obtained with an interleukin-1β (IL-1β)-blocking antibody in the CANTOS trial have validated the inflammatory paradigm of atherosclerosis in humans and demonstrated the potential utility of anti-inflammatory drugs in patients with atherosclerotic disease. However, CANTOS also highlighted the need for molecular strategies with improved selectivity and less side effects[4].

While anti-chemokine strategies such as antibodies or small molecule drugs (SMDs) have been established, targeting a specific chemokine/receptor axis remains challenging due to the promiscuity in the chemokine network[1,3,5,6]. In addition to antibodies and SMDs, soluble receptor-based approaches have proven as a powerful anti-cytokine strategy in inflammatory/immune diseases. For example, soluble tumor necrosis factor-receptor-1 (TNFR1)-based drugs are in clinical use for rheumatoid arthritis[7]. However, soluble receptor-based approaches are not established for chemokine receptors because of the discontinuous nature of the GPCR ectodomain topology.

Macrophage migration-inhibitory factor (MIF) is an evolutionarily conserved, multi-functional inflammatory mediator that is structurally distinct from other cytokines[8–12]. The MIF protein family also comprises D-dopachrome tautomerase (D-DT)/MIF-2[13]. MIF is an upstream regulator of the host innate and adaptive immune response, and, if dysregulated, is a key driver of acute and chronic inflammation, and cardiovascular diseases including atherosclerosis[8,9,11,12,14–18]. Atherosclerotic vascular inflammation from leukocyte recruitment to foam cell formation and advanced plaque remodeling is orchestrated by chemokines[1,3]. Examples are the classical chemokines CC-motif chemokine ligand 2 (CCL2) and CXC-motif chemokine ligand 1 (CXCL1)/CXCL8, but atypical chemokines (ACKs) that are structurally different from classical chemokines and yet interact with CKRs, have emerged as additional players in inflammation and atherogenesis[16,17]. Contrary to its eponymous name, MIF is recognized as a prominent ACK that enhances atherogenic leukocyte recruitment through non-cognate interactions with CXC-motif chemokine receptors type 2 (CXCR2) and 4 (CXCR4)[14,16,17]. Furthermore, MIF (and also MIF-2) are the sole ligands for the single-spanning type-II membrane protein CD74/invariant chain, through which they exert cardioprotective effects in the ischemic heart[15,16,19,20].

We recently elucidated the structural determinants of the binding interface between MIF and its receptors. Binding of MIF to CD74 involves MIF residues Pro-2, 80–87, and Tyr-100, while MIF binding to CXCR2 requires a pseudo-ELR motif, similar to CXCL8[19,21–23]. In contrast, the cognate CXCR4 ligand CXCL12/ stromal cell-derived factor-1alpha (SDF-1α) is an ELR-negative chemokine and the CXCL12/CXCR4 interface involves the receptor N-terminus and the RFFESH motif of CXCL12 at site 1 and the chemokine N-terminus and an intramembrane receptor groove at site 2. MIF binding to CXCR4 encompasses an extended N-like loop of MIF with contribution from Pro-2. Unlike for CXCL12, the MIF N-terminus around Pro-2 is rigid and likely unable to insert into the groove of CXCR4. On the side of CXCR4, segments of extracellular loop 1 (ECL1) and extracellular loop 2 (ECL2) but not extracellular loop 3 (ECL3) contribute to the MIF/CXCR4 interface[24,25].

We thus reasoned that designing CXCR4 ectodomain-derived peptides mimicking its interaction surface with MIF might be a promising approach to develop receptor-selective MIF inhibitors. Moreover, as the CXCL12/CXCR4 pathway exhibits critical homeostatic functions in resident arterial endothelial and smooth muscle cells and has a critical atheroprotective role[26,27], we aimed to generate CXCR4 mimics specific for MIF/CXCR4, while sparing CXCL12 pathways. Such mimics would be soluble chemokine receptor ectodomain-based inhibitors with receptor- and agonist-selective targeting properties. This approach would address current gaps in tailored chemokine-selective targeting strategies and receptor-specific MIF therapeutics in inflammatory and cardiovascular diseases.

Using rational peptide design, structure-activity relationships (SAR), and an array of biophysical methods, we here report on engineered CXCR4 ectodomain-derived peptide mimics that selectively bind to the atypical chemokine MIF but not to CXCL12. Signaling experiments, chemotaxis, foam cell formation, and leukocyte recruitment studies in vitro and in the atherosclerotic vasculature demonstrate that such mimics can act as agonist-specific anti-atherogenic compounds, blocking CXCR4-mediated atherogenic MIF activities, while sparing CXCL12 and protective MIF/CD74-dependent signaling in cardiomyocytes. We show that the CXCR4 mimic msR4M-L1 is not only enriched in atherosclerotic plaque tissue in a MIF-specific manner in mouse and human lesions, but functionally protects from lesion development and atherosclerotic inflammation in an atherogenic Apolipoprotein e-deficient (Apoe[−/−]) model in vivo.

## Results

**Designing chemokine-selective CXCR4 ectodomain mimics.** Our previous peptide array and SAR studies showed that residues 97–110 of ECL1 and 182–196 of ECL2 of the CXCR4 ectodomain contribute to the interface between MIF and CXCR4[25]. We reasoned that this could be a basis to engineer soluble MIF-binding CXCR4 mimics. Peptides ECL1[97–110] and ECL2[182–196] were synthesized by solid-phase peptide synthesis (SPPS) using Fmoc chemistry[28]. A synthetic linker was chosen based on the CXCR4 X-ray structures[28–30]. We designed and generated the conformationally constrained ectodomain mimic CXCR4-ECL1 [97–110]-6-Ahx-12-Ado-ECL2[182–196] (MIF-specific human CXCR4 mimic-ECL1[97–110]-6-Ahx-12-Ado-ECL2[182–196] or 'msR4M-L1'; Fig. 1a, b; Supplementary Fig. 1; Supplementary Table 1; Supplementary Fig. 2) that contained a 6-aminohexanoic acid (6-Ahx)/12-amino-dodecanoic acid (12-Ado) linker with a spacer length of 2.358 nm. 8-amino-3,6-dioxaoctanoic acid (O2Oc)/12-Ado was chosen as an alternative, more hydrophilic, linker ('msR4M-L2'; Supplementary Fig. 1; Supplementary Table 1; Supplementary Fig. 3). We also synthesized the non-linked loop peptides for comparison as well as variants of msR4M-L1 and -L2 that were additionally constrained by a disulfide bridge in the presence (msR4M-L1ox and -L2ox) or absence of the synthetic linker (msR4M-LS) (Supplementary Table 1). We did not include residues of the CXCR4 N-terminal, because this region has been implicated as a critical region contributing to the CXCL12/CXCR4 interface[29–31], which we wished to specifically exclude from our targeting strategy.

To determine whether the CXCR4 ectodomain mimics bind to MIF, we first applied fluorescence titration spectroscopy[32,33], measuring changes in fluorescence emission of Fluos-labeled ectodomain peptide upon titration against MIF or CXCL12. Conversely, Alexa-Fluor 488-labeled MIF (Alexa-MIF) was titrated against unlabeled ectodomain peptides. As determined by this method, msR4M-L1 exhibited high-affinity binding to MIF with an apparent (app.) $K_D \leq 40$ nM (app. $K_D$ Fluos-msR4M-L1/MIF = $40.7 \pm 4.0$ nM; app. $K_D$ msR4M-L1/Alexa-MIF = $31.1 \pm 16.6$ nM), whereas no binding to CXCL12 was

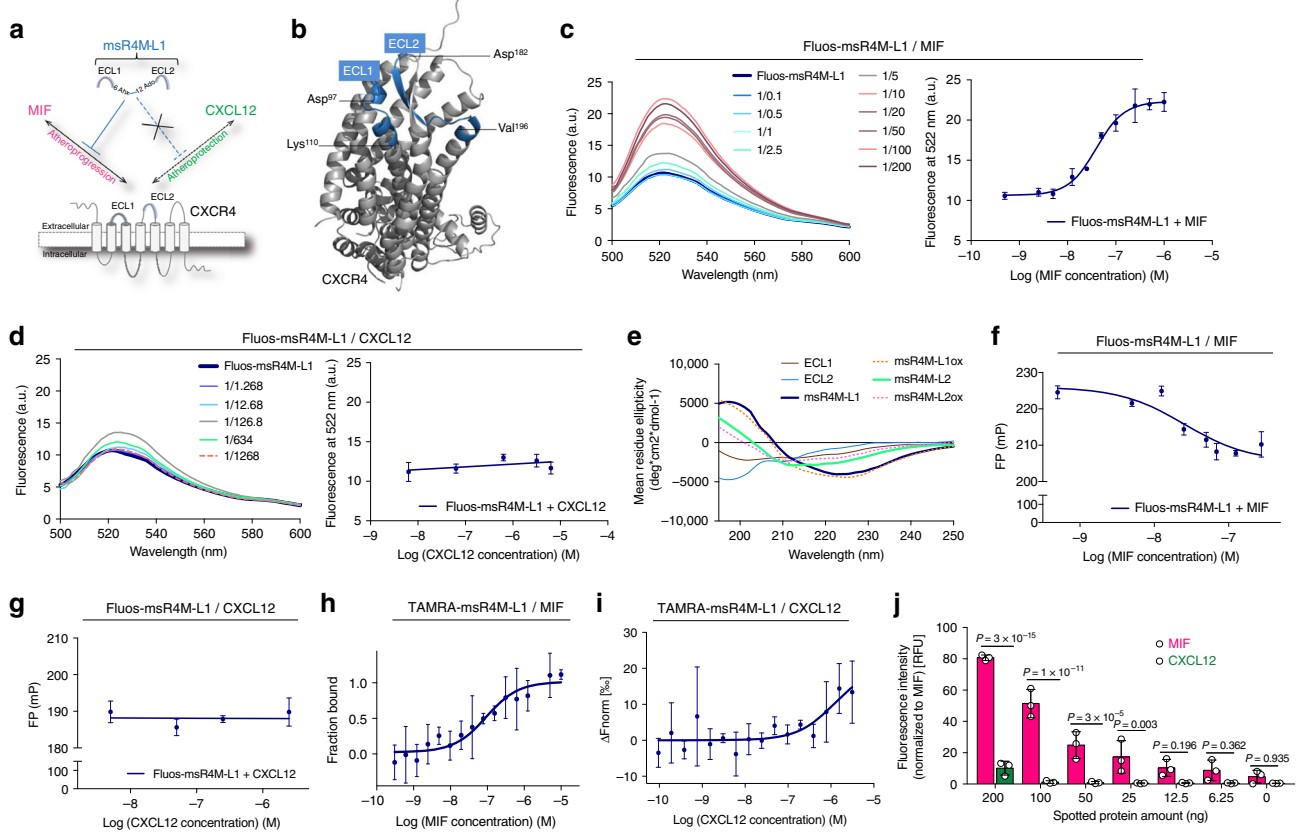

**Fig. 1 The CXCR4 ectodomain mimic msR4M-L1 selectively binds to MIF but not CXCL12. a** Schematic summarizing the design strategy to utilize extracellular loop moieties of CXCR4 to engineer a soluble mimic that binds MIF but not CXCL12. **b** Ribbon structure of human CXCR4 based on the crystal structure according to PDB code 4RWS[30]. Sequences of extracellular loops ECL1 and ECL2 that were found to interact with MIF according to peptide array mapping[25] are highlighted in blue, and the N- and C-terminal residues of the ECL1 and 2 peptides are indicated. **c** Nanomolar affinity binding of msR4M-L1 to MIF as determined by fluorescence spectroscopic titrations. Emission spectra of Fluos-msR4M-L1 alone (blue; 5 nM) and with increasing concentrations of MIF at indicated ratios are shown (left panel; representative titration); binding curve derived from the fluorescence emission at 522 nm (right panel). **d** msR4M-L1 does not bind to CXCL12. Same as **c** but titration performed with increasing concentrations of CXCL12. **e** Conformation of CXCR4 ectodomain mimics as determined by far-UV CD spectroscopy. Mean residue ellipticity plotted over the wavelength between 195 and 250 nm. **f**–**g** Nanomolar binding of msR4M-L1 to MIF (**f**) but not CXCL12 (**g**) as determined by fluorescence polarization (FP). The FP signal of 5 nM Fluos-msR4M-L1 (as mP) is plotted over varied ligand concentration as indicated. **h**, **i** Binding of msR4M-L1 to MIF (**h**) but not CXCL12 (**i**) as confirmed by microscale thermophoresis (MST). The fraction of chemokine bound or normalized fluorescence change (ΔFnorm) to 100 nM TAMRA-msR4M-L1 is plotted against increasing concentrations of MIF or CXCL12, respectively. **j** Binding analysis between TAMRA-msR4M-L1 and MIF versus CXCL12 as determined by dot blot titration. Quantification from three independent blots (see representative blot in Supplementary Fig. 8). Data in **c**–**d** (right panel) and **f**–**h** are reported as means ± SD from three independent experiments; data in **i** are means ± SD from five independent experiments. Statistical analysis (**j**) was performed with two-way ANOVA and Sidak correction. CXCR4, CXC motif chemokine receptor-4; msR4M-L1, MIF-specific CXCR4 mimic-L1; MIF, macrophage migration-inhibitory factor. Source data are provided as a Source Data file.

observed (app. $K_D > 6 \mu M$) (Table 1; Fig. 1c, d; Supplementary Fig. 4). msR4M-L2 bound to MIF with similar affinity and also lacked CXCL12 binding (app. $K_D$ Fluos-msR4M-L2/MIF = 18.6 ± 2.9 nM; app. $K_D$ msR4M-L2/Alexa-MIF = 40.5 ± 7.6 nM; app. $K_D$ Fluos-msR4M-L2/CXCL12 > 6 μM; Table 1; Supplementary Fig. 4). Thus, both mimics exhibited a >140-fold selectivity for MIF versus CXCL12. Interestingly, fluorescence titration spectroscopy measurements also indicated that additional conformational restriction of the mimics by disulfide bridging leads to an induction of CXCL12 binding, while high-affinity MIF binding was preserved in such analogs (Table 1; Supplementary Fig. 4). By contrast, the single, non-linked ECL peptides ECL1[97–110] and ECL2[182–196] only exhibited a medium-low binding affinity for MIF (app. $K_D$ ECL1/Alexa-MIF = 345.2 ± 79.4 nM; app. $K_D$ ECL2/Alexa-MIF = 2458 ± 1054 nM; Table 1; Supplementary Fig. 4).

Of note, circular dichroism (CD) spectroscopy indicated that especially msR4M-L1 is well-structured (Fig. 1e). CD

spectroscopy also confirmed our design strategy with appreciable conformational restriction introduced by the 6-Ahx-12-Ado linker of msR4M-L1, as a non-linked mixture of ECL1 and 2 exhibited mostly random coil conformation (Supplementary Fig. 5). Thus, the initial fluorescence and CD spectroscopic experiments suggested that msR4M-L1 (and -L2), but not the disulfide-bridged or single ECL loop variants, could represent promising CXCR4 mimics with high selectivity for MIF versus CXCL12. However, fluorescence spectroscopy also indicated that both msR4M-L1 and -L2 exhibit some degree of self-assembly propensity (app. $K_D$ Fluos-msR4M-L2/msR4M-L2 = 69.6 ± 61.9 nM versus Fluos-msR4M-L1/msR4M-L1 = 142.0 ± 48.9 nM; Supplementary Fig. 6). Although this was 2-fold lower for msR4M-L1 compared to -L2 and the concentrations of msR4M-L1 used in the fluorescence spectroscopic titrations were >25-fold lower than the app. $K_D$ of its self-assembly, we sought to support our findings by additional methods, as fluorescence intensity may be affected by several factors. First, we applied fluorescence polarization (FP),

**Table 1 Binding affinities between the CXCR4 ectodomain peptides and MIF versus CXCL12 as determined by fluorescence titration spectroscopy, fluorescence polarization (FP), and microscale thermophoresis (MST).**

| CXCR4 ectodomain (ECD) peptide | Fluos (TAMRA)-ECD peptide/MIF[a, b] (app. $K_D$ [nM][c]) | Alexa-MIF/ECD peptide[d] (app. $K_D$ [nM]) | Fluos (TAMRA)-ECD/CXCL12[e] (app. $K_D$ [nM]) |
|---|---|---|---|
| *Fluorescence spectroscopy* | | | |
| msR4M-L1 | 40.7 ± 4.0 | 31.1 ± 16.6 | >6340 |
| msR4M-L2 | 18.6 ± 2.9 | 40.5 ± 7.6 | >6340 |
| msR4M-L1ox | 28.9 ± 2.5 | 30.0 ± 6.3 | 84.6 ± 42.1 |
| msR4M-L2ox | 105.3 ± 44.9 | 59.6 ± 15.3 | 54.8 ± 10.3 |
| msR4M-LS | 6.9 ± 2.0 | n.d. | 17.4 ± 4.7 |
| ECL1 | n.d. | 345.2 ± 79.4 | n.d. |
| ECL2 | >5000 | 2458 ± 1054 | n.d. |
| *Fluorescence polarization (FP)* | | | |
| msR4M-L1 | 24.4 ± 5.3 | 10.6 ± 1.2 | >2500 |
| *Microscale thermophoresis (MST)* | | | |
| msR4M-L1 | 77.2 ± 37.1 | n.d. | >3125 |

[a]For fluorescence spectroscopy and FP, Fluos-labeled ECD peptides were used at a concentration of 5 nM
[b] For MST, TAMRA-labeled ECD peptide was used at a concentration of 100 nM
[c] Reported apparent $K_D$ values are means ± SD from 3 (fluorescence spectroscopy and FP) or 5 (MST) independent experiments and were calculated as described in Methods
[d] Alexa-MIF was used at a concentration of 10 nM
[e]Alexa-CXCL12 measurements were not pursued, because of the notion that Alexa labeling could interfere with the crucial residue Lys-1 of CXCL12[30] as well as other binding-relevant lysines. ECD, extracellular domain; app., apparent; n.d., not determined

which is a robust method for quantifying affinities of protein-protein interactions and has previously been applied to study MIF interactions[34]. Confirming the results obtained by fluorescence spectroscopy, incubation of Fluos-msR4M-L1 with increasing concentrations of MIF resulted in significant changes in FP and an app. $K_D$ in the low nanomolar range was determined (app. Fluos-msR4M-L1/MIF = 24.4 ± 5.3 nM; Fig. 1f). A similar app. $K_D$ was obtained for the Alexa-MIF/msR4M-L1 interaction (app. $K_D$ Alexa-MIF/msR4M-L1 = 10.6 ± 1.2 nM; Supplementary Fig. 7). In contrast, no change in FP was observed, when Fluos-msR4M-L1 was titrated with CXCL12 (app. $K_D$ > 2.5 µM; Fig. 1g). In addition to FP, the interactions between msR4M-L1 and MIF or CXCL12 were studied by microscale thermophoresis (MST). For the interaction between 5(6)-carboxytetramethylrhodamine (TAMRA)-msR4M-L1 and MIF an app. $K_D$ of 77.2 ± 37.1 nM was found (Fig. 1h). By contrast, the app. $K_D$ of the TAMRA-msR4M-L1/CXCL12 interaction was >3 µM (Fig. 1i). Thus, the FP and MST studies confirmed the findings of the fluorescence spectroscopic titrations (summarized in Table 1). Further evidence for the selectivity of the interaction between msR4M-L1 and MIF as compared with CXCL12 was obtained by a dot blot experiment, in which immobilized MIF and CXCL12 were probed on a dot blot membrane with TAMRA-msR4M-L1. MIF was readily detected in a concentration-dependent manner, whereas TAMRA-msR4M-L1 showed no binding to CXCL12 (Fig. 1j; Supplementary Fig. 8).

MIF residues 80–87 and Tyr-100 are specific determinants of the interaction between MIF and CD74, whereas Pro-2 of MIF contributes to both MIF/CD74 binding and the MIF/CXCR4 interface[21,22,25]. We therefore also applied FP and MST in a binding-competition approach to experimentally confirm that msR4M-L1 does not interfere with MIF binding to its receptor CD74, which mediates MIF's cardioprotective activity[15]. Applying FP, we determined an app. $K_D$ of 114.4 ± 47.0 nM for the interaction between Alexa-MIF and an HA-tagged soluble ectodomain of CD74 (HA-tagged sCD74(73–232)), overall in

line with the previously measured binding affinity between MIF and CD74[19]. Binding was not affected by co-incubation of MIF with msR4M-L1 (app. $K_D$ = 89.4 ± 55.3 nM; $P$ = ns) (Supplementary Fig. 9). MST experiments confirmed that binding between MIF and CD74 was not inhibited by msR4M-L1. Titration of Alexa-647-labeled MIF with increasing concentrations of HA-tagged sCD74(73–232) was not influenced by msR4M-L1 (app. $K_D$ Alexa-647-MIF/HA-sCD74(73–232) = 33.9 ± 5.0 nM; Alexa-647-MIF/HA-sCD74(73–232) + msR4M-L1 = 34.5 ± 13.1 nM; $P$ = ns; Supplementary Fig. 9).

Together, the data demonstrate that msR4M-L1, an engineered soluble CXCR4 ectodomain mimic, binds with high affinity to MIF, exhibiting binding selectivity for MIF versus the cognate ligand CXCL12, while not interfering with MIF/CD74 binding. This led us to prioritize msR4M-L1 for further analysis.

**Mapping and mutation of the MIF/msR4M-L1 binding region.** We mapped the binding region in MIF and msR4M-L1 by fragment approach and alanine-scanning (Fig. 2a, b). As our structure-activity studies on the MIF/CXCR4 interface had provided evidence for a role of the N-like loop[24,25], we started the mapping with a MIF peptide fragment spanning this region (Fig. 2a). Applying fluorescence spectroscopy, MIF peptide 38–80 was found to bind to msR4M-L1 with similar affinity as full-length MIF (app. $K_D$ Fluos-msR4M-L1/MIF[38–80] = 57.1 ± 7.8 nM; Fig. 2a,c; Supplementary Table 2; Supplementary Fig. 10). As the peptide lacks the 3D conformation of folded full-length MIF, this suggested that a locally-defined sequence was important for the interaction with the ectodomain mimic. Moreover, detailed mapping of the msR4M-L1/MIF binding region by analyzing various 14–30-meric MIF peptide fragments spanning regions within and outside of sequence 38–80, narrowed the core binding region in MIF to sequence 50–80 or 54–80 (app. $K_D$ Fluos-msR4M-L1/MIF[50–80] = 55.2 ± 9.9 nM; app. $K_D$ Fluos-msR4M-L1/MIF[54–80] = 70.6 ± 14.2 nM; Fig. 2d; Supplementary Table 2; Supplementary Fig. 10). On the other hand, peptide segment 6–23, which is outside the core region, or 62–80 which is situated at the far C-terminal end of the core, did not bind to MIF (app. $K_D$ Fluos-msR4M-L1/MIF[6–23] >20 µM; app. $K_D$ Fluos-msR4M-L1/MIF[62–80] >10 µM; Fig. 2e; Supplementary Table 2; Supplementary Fig. 10; Supplementary Fig. 11). Notably, the identified MIF core region bound to msR4M-L2 as well (app. $K_D$ Fluos-msR4M-L2/MIF[50–80] = 30.9 ± 20.4 nM; app. $K_D$ Fluos-msR4M-L2/MIF[54–80] = 52.9 ± 25.6 nM).

To further map the inhibitor binding region of MIF as well as the binding determinants of msR4M-L1 and verify the specificity of msR4M-L1/MIF complex formation, we next studied alanine mutants of both msR4M-L1 and MIF. Synthetic peptide arrays comprising 14–15-meric segments within msR4M-L1-related ECL1 and ECL2 sequences in combination with extensive alanine-scanning and array probing with biotinylated human MIF (Supplementary Table 1; Supplementary Fig. 12) guided us regarding the critical residues within msR4M-L1. Using single, double, and triple alanine mutants, sequences 102–108 and 188–196 containing several aromatic residues were identified as possible MIF binding core regions in the ECL1 and ECL2 parts of msR4M-L1, respectively. Accordingly, the aromatic Phe, Trp, and Tyr residues in position 102, 103, 104, 107, 189, 190, and 195 could be key determinants of msR4M-L1 binding to MIF (Supplementary Fig. 12). We applied fluorescence titration spectroscopy to study the MIF binding properties of a mutant of msR4M-L1, in which all of these aromatic residues were substituted by Ala, i.e. analog [A[102], A[103], A[104], A[107], A[189], A[190], A[195]]-msR4M-L1 or msR4M-L1(7xAla). In fact, disruption of the predicted core MIF binding site in msR4M-L1(7xAla) ablated its

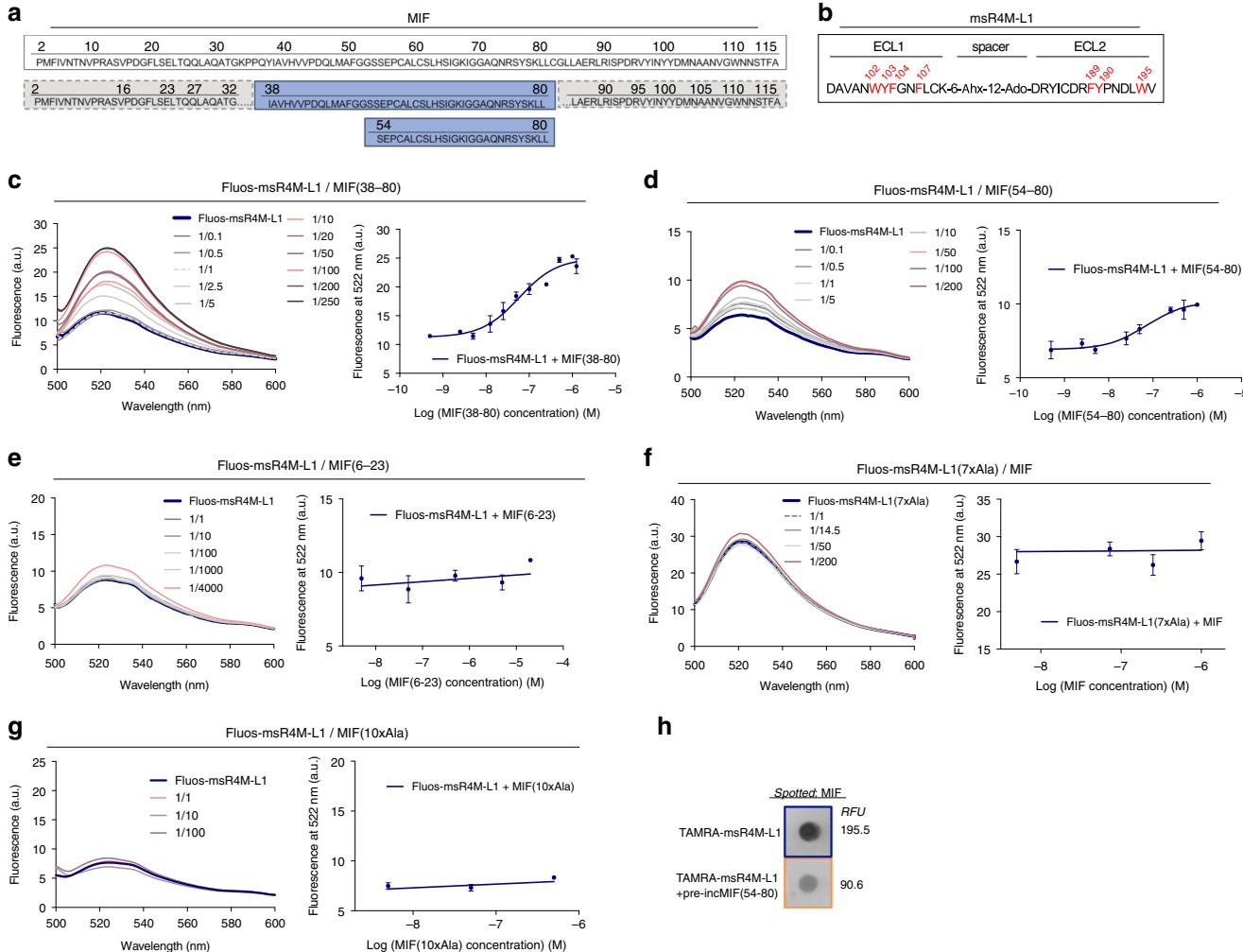

**Fig. 2 Mapping of the MIF/msR4M-L1 core binding region and complex disruption by mutations. a** Amino acid sequence of human MIF (boxed, top). The msR4M-L1 binding core region of MIF (sequence 38–80 and 54–80) is indicated in blue, while non-binding stretches are in gray (bottom). **b** Sequence of msR4M-L1. Aromatic residues identified by peptide array to be critical for MIF binding are highlighted in red. **c–e** Nanomolar affinity binding of msR4M-L1 to MIF(38–80) (**c**) and MIF(54–80) (**d**), but not MIF(6–23) (**e**), as determined by fluorescence spectroscopy. Emission spectra of Fluos-msR4M-L1 alone (blue; 5 nM) and with increasing concentrations of MIF(38–80) (**c**), MIF(54–80) (**d**), and MIF(6–23) (**e**) (left panels; representative titrations); binding curves derived from the fluorescence emission at 522 nm (right panels). **f–g** Binding of msR4M-L1 to MIF is blunted when aromatic residues in msR4M-L1 are substituted by Ala in analog msR4M-L1(7xAla) (**f**) or when N-loop residues in MIF are mutated to Ala in MIF(10xAla) (**g**). Fluorescence spectroscopy and binding curve as in **c–e**. Data in right panels of **c–g** are means ± SD from three independent experiments. **h** Dot blot shows that binding of TAMRA-msR4M-L1 to spotted MIF is attenuated by MIF(54–80). 400 ng spotted MIF was probed with TAMRA-msR4M-L1 +/− 2-fold molar excess of MIF(54–80); RFU, relative fluorescent units. The blot shown is one of three dot blots performed. msR4M-L1, MIF-specific CXCR4 mimic-L1; MIF, macrophage migration-inhibitory factor. Source data are provided as a Source Data file.

binding to MIF (Fig. 2f; app. $K_D$ Fluos-msR4M-L1(7xAla)/MIF > 1 µM). As alanine substitutions of Phe-104 and Phe-107 caused the strongest observed reductions in spot intensity, we also determined whether the substitution of only these two residues would affect the binding capacity of msR4M-L1. In fact, peptide [A104, A107]-msR4M-L1 or msR4M-L1(2xAla) displayed substantially reduced binding to MIF (Supplementary Fig. 13; app. $K_D$ Fluos-msR4M-L1(2xAla)/MIF > 1 µM). In line with the binding data, CD spectroscopy showed that msR4M-L1(7xAla) and msR4M-L1(2xAla) fully and partially, respectively, lost the well-defined structure of msR4M-L1 and their spectra were consistent with large portions of random coil structure (Supplementary Fig. 13). We also performed an inverse mutation experiment, in which residues 47–56 within the MIF binding core were substituted by alanine by PCR mutagenesis. Mutant [A47–56]-MIF or MIF(10xAla) only retained a minimal binding

activity to msR4M-L1 as determined by fluorescence titration of Fluos-msR4M-L1 and this mutant (Fig. 2g; app. $K_D$ Fluos-msR4M-L1/MIF(10xAla) >500 nM). Lastly, we performed a dot blot-based competition experiment, in which the capacity of peptide MIF[54–80] (representing the core binding region of MIF) to compete with the binding between MIF and TAMRA-msR4M-L1 was tested. When TAMRA-msR4M-L1 was co-incubated with MIF[54–80], binding to immobilized MIF was markedly reduced (Fig. 2h).

Together, these data further confirmed the specificity of msR4M-L1/MIF complex formation. They also indicate that the aromatic residues in msR4M-L1, in particular Phe-104 and Phe-107, play a critical role for the binding to MIF and that the msR4M binding determinants of MIF are mainly located in MIF sequence stretch 38–80, or even 54–80, consistent with a role of the N-like loop[25].

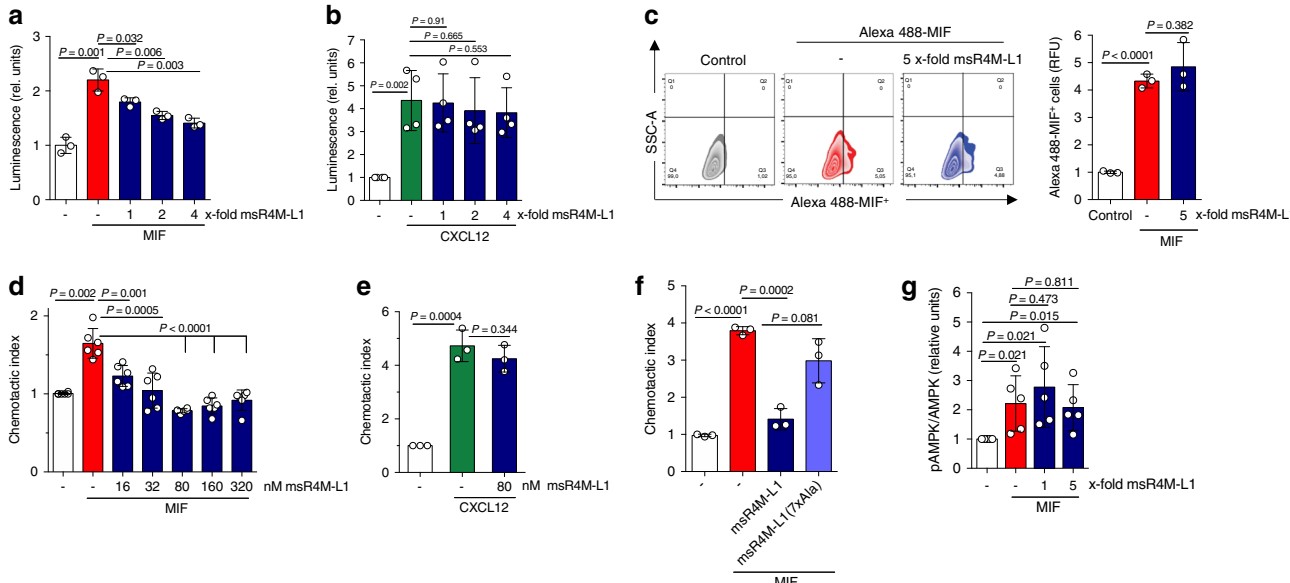

**Fig. 3 msR4M-L1 selectively inhibits MIF-triggered CXCR4 activity, but spares the MIF/CD74 axis. a**, **b** MIF (**a**) but not CXCL12 (**b**) binding to and signaling through human CXCR4 in an *S. cerevisiae* system is attenuated by msR4M-L1 in a concentration-dependent manner. The molar excess of competing msR4M-L1 over MIF or CXCL12 is indicated. CXCR4 binding/signaling is read out by LacZ reporter-driven luminescence. **c** A 5-fold molar excess of msR4M-L1 does not interfere with binding of Alexa 488-MIF to CD74 expressed on HEK293-CD74 transfectants as measured by flow cytometry. Left, shift of CD74 transfectants following Alexa 488-MIF binding (control indicates background); right, quantification of three independent experiments. **d**, **e** Chemotactic migration (Transwell) of primary mouse spleen B lymphocytes elicited by 16 nM MIF (**d**) or CXCL12 (**e**) as chemoattractant and inhibitory effect of msR4M-L1. msR4M-L1 dose-dependently inhibits MIF-mediated chemotaxis (**d**), but the optimal inhibitory dose of 80 nM does not affect CXCL12-elicited chemotaxis (**e**). **f** msR4M-L1 analog msR4M-L1(7xAla) does not inhibit MIF-mediated chemotaxis. msR4M-L1(7xAla) was applied at a concentration of 80 nM. **g** msR4M-L1 does not interfere with MIF-triggered AMPK signaling in the human cardiomyocyte cell line HCM. MIF was applied at a concentration of 16 nM; msR4M-L1 added at 1- and 5-fold excess over MIF. AMPK signaling was measured using Western blot of HCM lysates developed against pAMPK and total AMPK. The densitometric ratio of pAMPK/AMPK indicates signaling intensity. Data are reported as means ± SD of $n = 3$ (**a**); $n = 4$ (**b**); $n = 3$ (**c**, right panel); $n = 6$ (**d**); $n = 3$ (**e–f**); and $n = 5$ (**g**) independent biological experiments. Statistical analysis was performed with unpaired two-tailed T-test. CXCR4, CXC motif chemokine receptor-4; msR4M-L1, MIF-specific CXCR4 mimic-L1; MIF, macrophage migration-inhibitory factor. Source data are provided as a Source Data file.

**Ectodomain mimics selectively block MIF/CXCR4 activity**. To scrutinize whether selective msR4M-L1/MIF binding correlates with inhibition of MIF-triggered inflammatory and atherogenic effects, we first examined whether msR4M-L1 interfered with MIF/CXCR4-specific cell signaling. We took advantage of a yeast strain that expresses human CXCR4 and specifically measures agonist-mediated activation of CXCR4 via a reporter plasmid[25]. Confirming previous data[25], MIF triggered CXCR4-mediated signaling, but co-incubation of MIF with msR4M-L1 blocked the β-galactosidase reporter signal in a concentration-dependent manner (Fig. 3a). In contrast, CXCL12/CXCR4-elicited signaling remained unaffected by msR4M-L1 (Fig. 3b). This suggested that msR4M-L1 specifically blocks MIF/CXCR4-driven cell signaling responses.

Receptor signaling analysis in our yeast system is limited to GPCRs and not amenable to the non-GPCR receptor CD74. The FP and MST experiments with sCD74 indicated that msR4M-L1 does not affect MIF/sCD74 binding (Supplementary Fig. 9), but to verify that msR4M-L1 also does not interfere with the MIF/CD74 axis in a cell-based system, we transfected HEK293 cells with a construct driving CD74 surface expression[14]. Alexa-MIF cell-surface binding as measured by flow cytometry was elevated in a CD74-dependent manner. Of note, co-incubation of Alexa-MIF with a 5-fold molar excess of msR4M-L1 did not reduce surface binding of Alexa-MIF (Fig. 3c). Together, the yeast-CXCR4 and HEK293-CD74 transfectant data showed that msR4M-L1 blocks the interaction between MIF and cell-surface-expressed CXCR4, whereas binding to cell-surface CD74 is not affected.

We next asked whether msR4M-L1 also selectively inhibits MIF responses in mammalian cell systems expressing endogenous CXCR4. B lymphocytes express substantial levels of CXCR4 and MIF has been shown to trigger murine B-cell chemotaxis in a CXCR4-dependent manner[35]. Human and murine MIF share 90% amino acid identity and there is a high degree of cross-species receptor activity[36]. There also is a high degree of sequence identity between human and murine MIF in the binding region for msR4M-L1 (MIF(38–80): 86%; MIF(54–80): 89%), and by applying dot blot titration, we confirmed that msR4M-L1 binds equally well to human and mouse MIF (Supplementary Fig. 14). We then subjected primary splenic B cells to MIF-triggered chemotaxis using the Transwell system. When co-incubated with MIF, msR4M-L1 (as well as msR4M-L2; Supplementary Fig. 15) fully blocked MIF-mediated B-cell chemotaxis in a concentration-dependent manner with maximal inhibition seen at a 5-fold molar excess (Fig. 3e) and an IC$_{50}$ in the range of 10–15 nM (Supplementary Fig. 15). In contrast, msR4M-L1 was unable to inhibit chemotaxis elicited by CXCL12 (Fig. 3f). Of note, analog msR4M-L1(7xAla) with seven binding-determining residues substituted by Ala failed to inhibit MIF-elicited chemotaxis (Fig. 3f), while analog msR4M-L1(2xAla) with only two Ala substitutions retained partial chemotaxis-inhibitory activity (Supplementary Fig. 16), confirming the specificity of binding inhibition in a relevant cell-based assay. Thus, the yeast signaling and B-cell migration data suggested that msR4M-L1 potently and selectively interferes with MIF/CXCR4-mediated cell responses.

MIF is a pro-atherogenic cytokine, but also has context-dependent 'local' protective activity on cardiomyocytes[14–17].

Before further evaluating the translational potential of our findings, we wished to exclude that msR4M-L1 interferes with protective MIF/CD74-mediated signaling in cardiomyocytes. Primary human cardiomyocytes (HCM; expressing CD74, Supplementary Fig. 17) were incubated with MIF in the presence or absence of msR4M-L1. We then analyzed phosphorylated AMP kinase (pAMPK) levels, a correlate of protective MIF/CD74-mediated signaling. As demonstrated previously[15], MIF upregulated pAMKP levels, but this effect was not attenuated by msR4M-L1 (Fig. 3g; Supplementary Fig. 17), confirming that the CXCR4 mimic does not cross-affect MIF activities through CD74.

**msR4M-L1 blocks atherogenic MIF activity in vitro/ex vivo**. MIF is a driver of atherogenic monocyte activity and inhibition of monocyte-dependent atherogenic inflammation is a preferred strategy to limit atherosclerotic lesion formation. As both MIF and CXCR4 have previously been linked to macrophage foam cell formation[37,38], we tested the effect of msR4M-L1 on MIF-triggered uptake of fluorescently labeled oxidized low-density lipoprotein (DiI-oxLDL) by human macrophages derived from peripheral blood mononuclear cells (PBMCs). MIF doubled foam cell formation in this setting and this effect was blocked by the pharmacological inhibitor AMD3100 (Supplementary Fig. 18), verifying CXCR4 dependency. Notably, msR4M-L1 dose-dependently inhibited MIF-mediated DiI-oxLDL uptake, with complete inhibition seen at a 3-fold molar excess of msR4M-L1 over MIF (Fig. 4a, b). The uptake of oxLDL by macrophages is mediated by scavenger receptors such as CD36, but studies in *Cd36/Sra* double knockout mice suggest a role for additional pathways[39].

As recent evidence suggested a contribution of native LDL uptake to macrophage foam cell formation[40] and as macrophage-expressed CXCR4 promotes this process in a MIF/CXCR4- but not CXCL12/CXCR4- specific manner[41], we next tested the capacity of msR4M-L1 to inhibit MIF-triggered uptake of fluorescently labeled native LDL (DiI-LDL). Confirming previous findings[41], uptake of DiI-LDL by human macrophages was enhanced by MIF in a CXCR4-dependent manner (Fig. 4c). Of note, msR4M-L1 dose-dependently inhibited MIF-mediated DiI-LDL uptake (Fig. 4c; Supplementary Fig. 19). Specificity of the inhibitory effect was further confirmed by msR4M-L1(7xAla) and msR4M-L1(2xAla), which failed to inhibit MIF-mediated DiI-LDL uptake, as predicted from the binding and chemotaxis experiments (Fig. 4d; Supplementary Fig. 16). The assay also appeared suitable to compare the inhibitory capacity of msR4M-L1 with that of established MIF inhibitors, i.e. the neutralizing anti-MIF monoclonal antibody (mAb) NIH/IIID.9 and the small molecule inhibitor ISO-1. The inhibitory capacity of msR4M-L1 was slightly better than that of ISO-1, but lower than that of NIH/IIID.9 (Fig. 4e). This notion was confirmed by comparing the binding affinity between msR4M-L1 and MIF with those of ISO-1 and anti-MIF mAbs. Neutralizing mAbs such as NIH/IIID.9 or BAX01 bind human or mouse MIF with a $K_D$ of 1–2 nM[16,42]; binding is thus more affine, albeit within a comparable nanomolar range, than that between MIF and msR4M-L1 ($\leq$40 nM). The $K_D$ for the MIF/ISO-1 interaction has not been reported, but the IC$_{50}$ value for MIF/CD74 binding is 10 $\mu$M[16,43]. Using fluorescence spectroscopic titration, we determined the $K_D$ between ISO-1 and MIF to be 14.4 $\pm$ 4.4 $\mu$M (Supplementary Fig. 20). Thus, while msR4M-L1 is superior to ISO-1 and NIH/IIID.9 in being receptor-selective, its inhibitory capacity and binding affinity are between that of small molecule inhibitors and anti-MIF mAbs.

We next tested the potency of msR4M-L1 towards MIF-elicited three-dimensional (3D) chemotaxis of PBMCs. We applied 3D-chemotaxis methodology and assessed single-cell migration tracks

via time-lapse microscopy. msR4M-L1 dose-dependently attenuated MIF-triggered motility of human monocytes as quantified by forward migration index. The pro-migratory effect of MIF was already ablated by a 2-fold molar excess of msR4M-L1 (Fig. 4f, g, i). By contrast, the CXCL12-induced cell motility response remained unaffected (Fig. 4h, j).

A major atherogenic process promoted by MIF is its effect on leukocyte adhesion in the atherosclerotic vasculature, an activity involving engagement of CXCR4[14]. To determine whether this function of MIF can be attenuated by CXCR4 mimics, MIF-triggered adhesion of MonoMac-6 monocytes on human aortic endothelial (HAoEC) monolayers under static conditions was assessed in the presence or absence of msR4M-L1. Figure 5a shows that msR4M-L1 blocked the pro-adhesion effect of MIF. Atherosclerosis is characterized by low-grade chronic inflammation in the vasculature with elevated levels of inflammatory cytokines such as TNF-α. To account for this situation, we next performed static adhesion on HAoECs prestimulated with TNF-α. MIF also increased MonoMac-6 adhesion on TNF-stimulated HAoECs, although the effect was less pronounced than in resting HAoECs, and msR4M-L1 abrogated the effect of MIF (Fig. 5b). To mimic physiological flow conditions, we next tested the inhibitory capacity of msR4M-L1 in a flow adhesion setting applying a shear stress of 1.5 dyn/cm$^2$ to an HAoEC monolayer super-fused with MonoMac6. Figure 5c shows that msR4M-L1 inhibited monocyte adhesion also under these hydrodynamic flow conditions.

Before studying atherogenic leukocyte recruitment effects in more pathogenically relevant ex vivo and in vivo settings, we wished to determine whether msR4M-L1 localizes to atherosclerotic plaque tissue. We stained plaque sections obtained from aortic root and brachiocephalic artery of atherogenic *Ldlr*$^{-/-}$ and *Apoe*$^{-/-}$ mice, respectively, with Fluos-msR4M-L1 to detect plaque binding of the CXCR4 mimic. Fluos-msR4M-L1 positivity was significantly more pronounced in plaque tissue from *Mif*-proficient atherogenic *Ldlr*$^{-/-}$ or *Apoe*$^{-/-}$ mice when compared to sections from *Mif*-deficient *Ldlr*$^{-/-}$ *Mif*$^{-/-}$ (Fig. 5d, e) or *Apoe*$^{-/-}$ *Mif*$^{-/-}$ (Supplementary Fig. 21) mice, respectively. This showed that Fluos-msR4M-L1, similar to anti-MIF antibody, was capable of binding to plaque-associated MIF. We next tested whether also in vivo administered Fluos-msR4M-L1 would localize to atherosclerotic plaque. Three days before vessel preparation, we intraperitoneally injected Fluos-msR4M-L1 into atherogenic *Apoe*$^{-/-}$ mice. Multiphoton laser-scanning microscopy (MPM) analysis of whole-mount carotid arteries from these mice visualized by second harmonic generation (SHG) and fluorescein detection revealed that Fluos-msR4M-L1 was markedly enriched in intimal plaque areas (Fluos-msR4M-L1 positivity = 26.8 $\pm$ 1.4%; Fig. 5f, g). A similar Fluos-msR4M-L1-positive area was detected in the aortic root (25.0 $\pm$ 1.9%; Fig. 5g, h), suggesting together that the CXCR4 mimic is enriched, at least partially, in atherosclerotic plaques in vivo.

To determine the functional consequence of this finding, we studied leukocyte recruitment in ex vivo mounted atherogenic carotid arteries using MPM. We injected mice with msR4M-L1 three days before vessel preparation and visualization of in situ adhering msR4M-L1- versus vehicle-exposed fluorescently labeled bone marrow-derived leukocytes in the vasculature under physiological flow conditions (Fig. 5i). In fact, the mimic significantly reduced the number of adhering leukocytes (Fig. 5j–l; Supplementary Movie 1).

Together, these findings suggested that msR4M-L1 localizes to atherosclerotic plaque tissue in a MIF-specific manner and inhibits MIF-mediated atherogenic leukocyte recruitment by interfering with chemotactic migration and arterial adhesion.

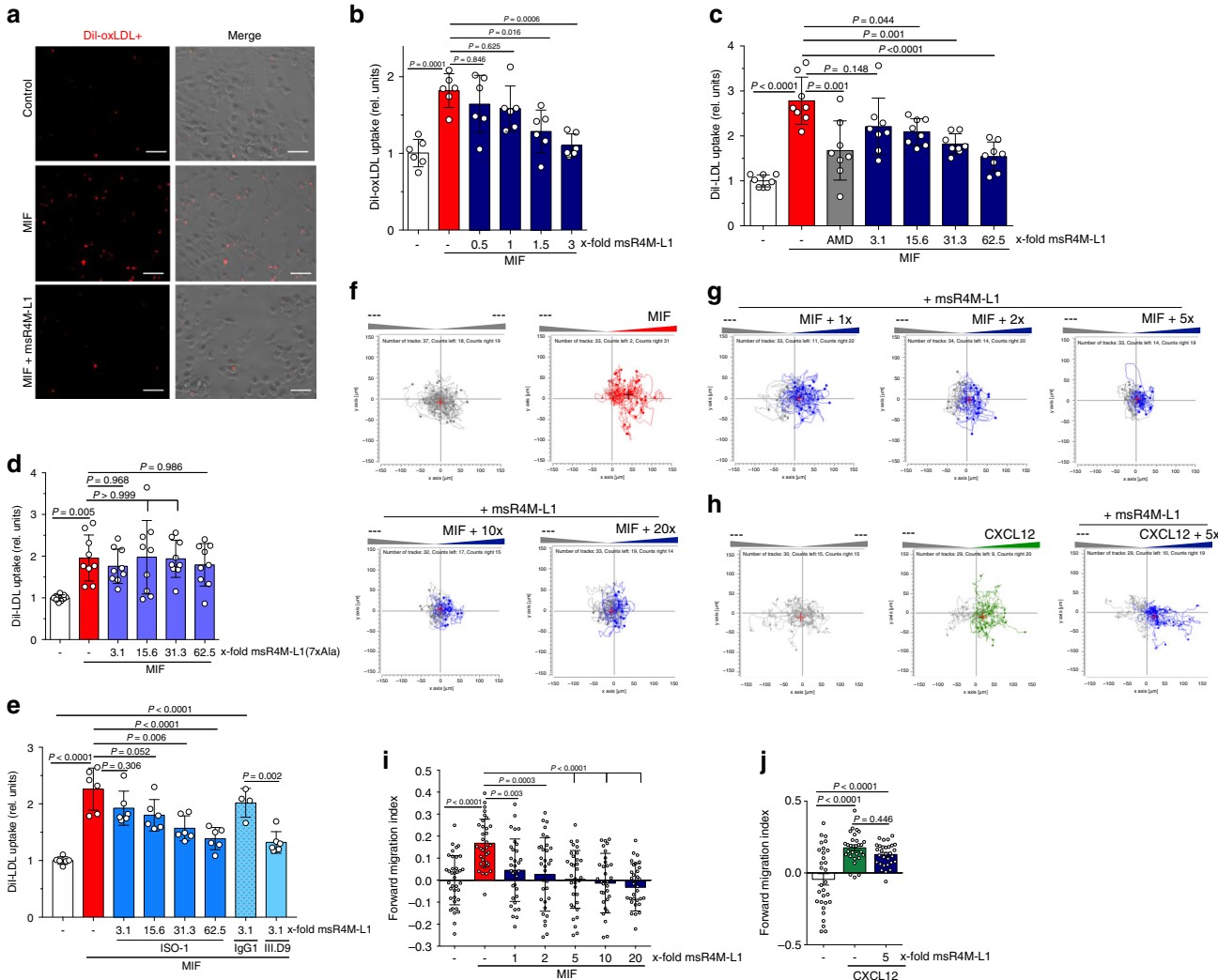

**Fig. 4 msR4M-L1 specifically inhibits MIF- but not CXCL12-elicited atherogenic monocyte activities. a**, **b** MIF-mediated DiI-oxLDL uptake in primary human monocyte-derived macrophages is dose-dependently inhibited by msR4M-L1 (indicated as molar excess over MIF). MIF was applied at a concentration of 80 nM. **a** Representative images of DiI-oxLDL-positive cells; **b** quantification (three-times-two independent experiments; 9 fields-of-view each). **c**, **d** MIF-specific DiI-LDL uptake in primary human monocyte-derived macrophages is dose-dependently inhibited by msR4M-L1 (indicated as molar excess over MIF) (**c**), but not by the MIF binding-dead analog of msR4M-L1, msR4M-L1(7xAla) (**d**). MIF was applied at a concentration of 80 nM. Quantification (four-times-two or three-times-two plus one-time-three, respectively, independent experiments; 9 fields-of-view each). AMD3100 (AMD) was used to verify CXCR4 dependence of the MIF effect. **e** Same as in **c**, **d**, except that the small molecule inhibitor ISO-1 and neutralizing MIF antibody NIH/IIID.9 were used instead of msR4M-L1 (three-times-two independent experiments; 9 fields-of-view each; isotype control antibody IgG1: two-times-two). **f**, **g** Representative experiment demonstrating that msR4M-L1 inhibits MIF-elicited (red tracks) 3D chemotaxis of human monocytes as assessed by live-microscopic imaging of single-cell migration tracks in x/y direction in μm. Increasing concentrations of msR4M-L1 (blue tracks, molar excess over MIF) as indicated; unstimulated control (gray tracks) indicates random motility. **i** Quantification of **f**, **g**; the migration tracks of 32–37 randomly selected cells per treatment group were recorded and the forward migration index plotted; the experiment shown is one of three independent experiments with monocytes from different donors. **h** A 5-fold molar excess of msR4M-L1 does not affect 3D human monocyte migration elicited by CXCL12; **j** quantification of **h**; the migration tracks of 29–30 randomly selected cells per treatment group were recorded and the forward migration index plotted; the experiment shown is one of two independent experiments with monocytes from different donors. Data in **b**–**e**, **i**, and **j** are reported as means ± SD. Statistical analysis was performed with one-way ANOVA with Tukey's multiple comparisons test or Kruskal–Wallis with Dunn's multiple comparisons test. The scale bar in **a** is: 50 μm. CXCR4, CXC motif chemokine receptor-4; msR4M-L1, MIF-specific CXCR4 mimic-L1; MIF, macrophage migration-inhibitory factor. Source data are provided as a Source Data file.

**The engineered CXCR4 mimic reduces atherosclerosis in vivo.** Peptides are sensitive to proteolysis by plasma proteases and clearance. Thus, before testing the potential therapeutic utility of msR4M-L1 in vivo, we examined its proteolytic stability and assessed its pharmacokinetic (PK) properties using fluorescently and biotin-labeled analogs of the mimic. TAMRA-msR4M-L1 was incubated with mouse plasma for up to 48 h. SDS-PAGE analysis and red fluorescence imaging (Fig. 6a) and densitometric

quantification of the TAMRA-msR4M-L1 band (Supplementary Fig. 22) revealed that approximately 30% of intact TAMRA-msR4M-L1 could be recovered upon 24–48 h of plasma exposure. A similar result was obtained, when Biotin-6-Ahx-msR4M-L1 was incubated with human plasma isolated from the blood of healthy donors (Supplementary Fig. 22), together indicating that this peptide was reasonably stable in mouse and human plasma. To assess the stability of msR4M-L1 in vivo, we i.p.-injected

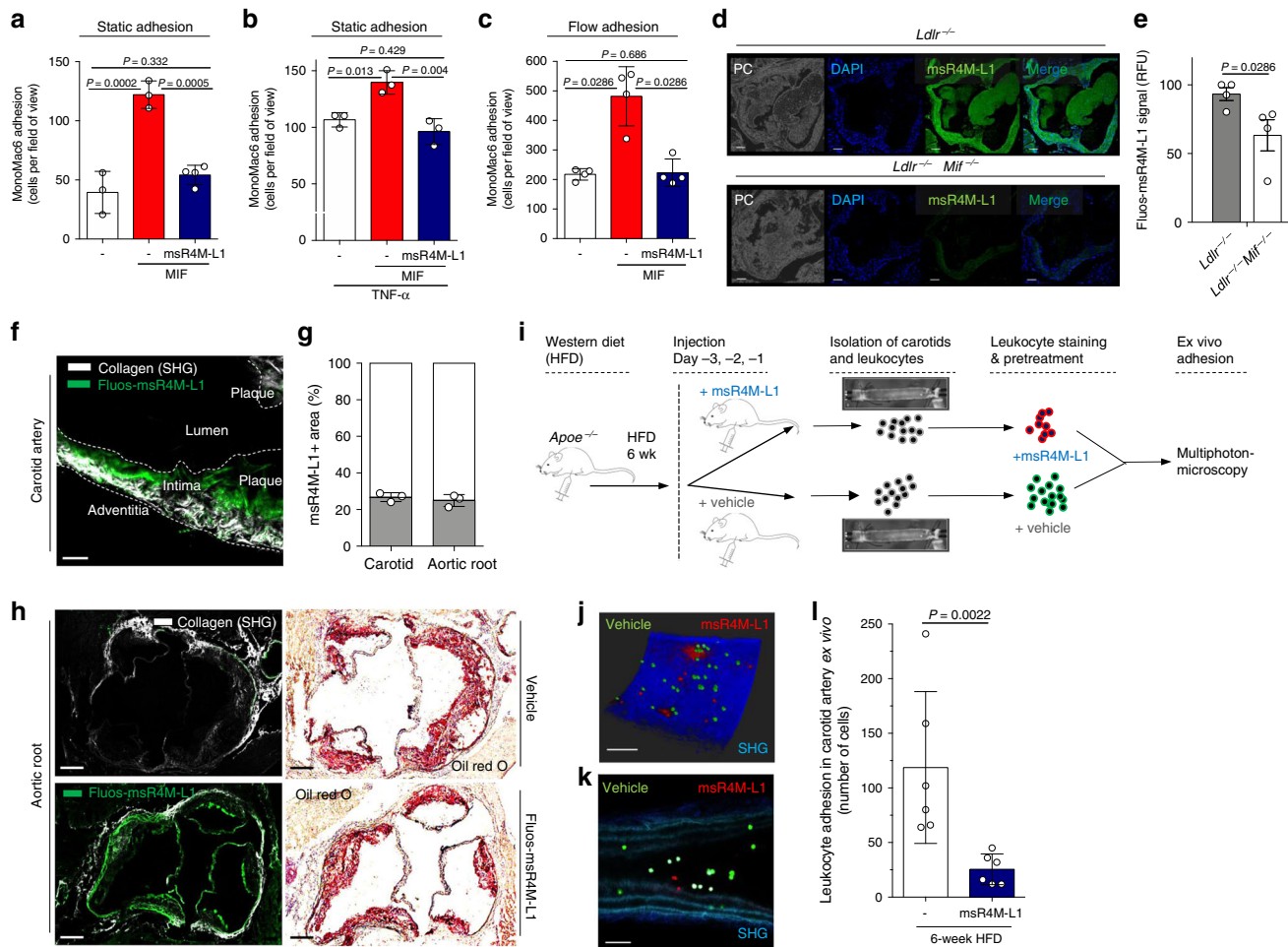

**Fig. 5 msR4M-L1 localizes to atherosclerotic plaques in a MIF-specific way and inhibits atherogenic leukocyte arrest. a, b** MIF-induced static adhesion of MonoMac-6 monocytes to HAoECs is ablated by msR4M-L1. **a** Resting HAoECs. **b** As in **a**, except that HAoECs were pre-incubated with TNF-α. Quantification based on 3 experiments with 10 independent fields-of-view each. **c** MIF-induced adhesion of MonoMac-6 monocytes to HAoECs under flow conditions (1.5 dyn/cm$^2$) is ablated by msR4M-L1. One of two independent experiments with four analyses each. **d, e** Fluos-msR4M-L1 stains aortic root sections from atherogenic $Ldlr^{-/-}$ mice on HFD in a MIF-specific manner (comparison between $Ldlr^{-/-}$ and $Ldlr^{-/-}Mif^{-/-}$ mice). **d** Representative images (PC, phase contrast; DAPI, nuclei); **e** quantification (relative fluorescence units) from two independent experiments with two animals each indicates MIF-specific staining. **f, g** Multiphoton laser-scanning microscopy image (**f**) of a carotid artery prepared from a hyperlipidemic $Apoe^{-/-}$ mouse, showing that in vivo administered Fluos-msR4M-L1 localizes to atherosclerotic plaques. Vessel visualization by second harmonic generation (SHG). **g** Quantification of the Fluos-msR4M-L1-positive area (means of $n = 3$ sections) as percentage of vessel target-area. **h** (and **g**) Same as **f, g**, except that aortic root was prepared (green in upper panel is autofluorescence). Quantification in **g** indicates ORO-positive target-area. **i–l** msR4M-L1 inhibits leukocyte adhesion in atherogenic carotid arteries under flow as analyzed by MPM. **i** Schematic summarizing the ex vivo leukocyte adhesion experiment. msR4M-L1 or vehicle was injected before vessel harvest; flushed leukocytes are stained in red (msR4M-L1; CMPTX) or green (vehicle; CMFDA). **j** Representative image of a carotid artery showing that pre-treatment with msR4M-L1 (red) leads to reduced luminal leukocyte adhesion compared to control (green), imaged by 3D reconstruction after Z-stacking (0.8–1.5 μm) (blue: SHG). **k** Still image of a z sectioning video scan (single field of view) through the artery (morphology revealed by SHG: collagen, dark blue; elastin, light blue). **l** Quantification of 5–6 independent carotid arteries per group. Luminally-adhering cell numbers are plotted. Scale bars: **d**, 50 μm; **f**, 100 μm; **h**, 100 μm; **j–k**, 100 μm. Data in **a**, **b**, **c**, **e**, **g** and **l** are reported as means ± SD. Statistical analysis was performed with one-way ANOVA with Tukey's multiple comparisons test or two-tailed Mann–Whitney test as appropriate. The vessel icon in Fig. 5i was used with permission from S. Karger AG (Copyright © 2006, © 2007 S. Karger AG, Basel[78]). msR4M-L1, MIF-specific CXCR mimic-L1; MIF, macrophage migration-inhibitory factor. Source data are provided as a Source Data file.

TAMRA-msR4M-L1 into 8-week-old C57BL/6 mice (50 μg per mouse or 2.5 mg kg$^{-1}$) and obtained plasma at intervals between 0 and 48 h. SDS-PAGE analysis and fluorescence imaging of the TAMRA-msR4M-L1 band revealed an appreciable in vivo stability of the ectodomain mimic (Fig. 6b). Capitalizing on a TAMRA-msR4M-L1 dose curve recorded in mouse plasma, we estimated the plasma concentration of msR4M-L1. One hour following i.p. administration, a plasma level of 180 ng mL$^{-1}$ was estimated, which declined by about 50% over the 48 h-time-course (Fig. 6b; Supplementary Fig. 23).

To examine the therapeutic capacity of the CXCR4 mimic, we employed an established in vivo mouse model of early atherosclerosis, in which lesions develop in aortic root and arch over a 4–5-week time course of HFD[44]. $Apoe^{-/-}$ mice received msR4M-L1 (50 μg per mouse or 2.5 mg kg$^{-1}$ i.p., three times per week, dosing instructed by the above-determined in vivo stability) or vehicle treatment in parallel to HFD for 4.5 weeks (Fig. 6c). We did not observe any significant effect of msR4M-L1 administration on body weight, plasma total cholesterol, triglyceride levels, or blood leukocyte counts, although lymphocytes showed a trend

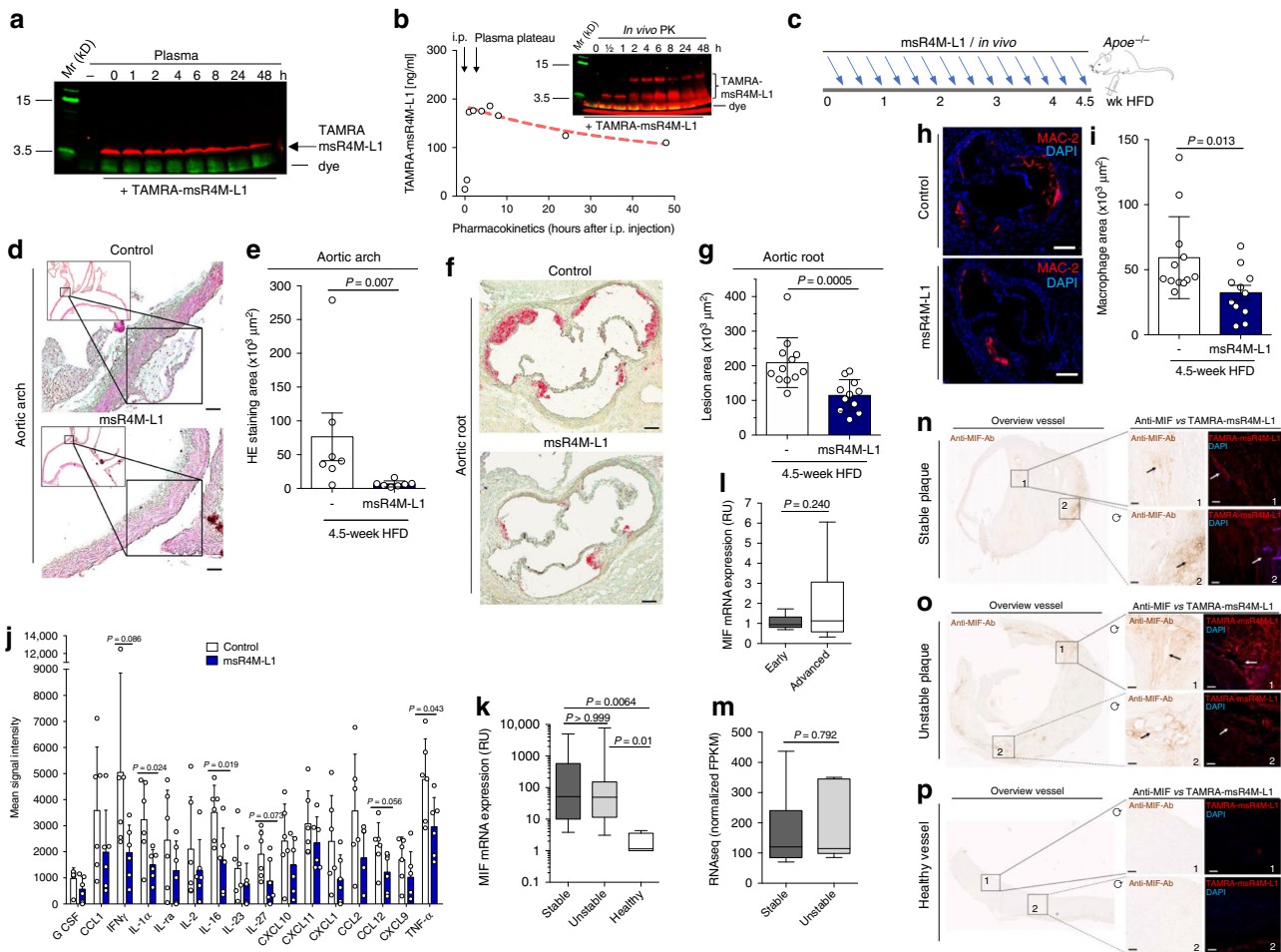

**Fig. 6 msR4M-L1 inhibits atherosclerosis in vivo and msR4M-L1-based staining parallels MIF expression in human plaques. a** SDS-PAGE imaging of TAMRA-msR4M-L1 mouse plasma incubations verifies proteolytic stability (red bands, TAMRA-msR4M-L1; green, running-dye). **b** In vivo stability of msR4M-L1. TAMRA-msR4M-L1 i.p.-injected into C57BL/6 mice and plasma collected as indicated. The pharmacokinetic is derived from SDS-PAGE TAMRA-msR4M-L1 bands (inset) applying a TAMRA-msR4M-L1 calibration curve (Supplementary Fig. 23). **c** Schematic showing the in vivo injection regimen in atherogenic *Apoe*$^{-/-}$ mice. **d, e** msR4M-L1 treatment reduces atherosclerotic lesions in aortic arch. Representative images (**d**) and quantification (**e**) of HE-stained sections from 7 msR4M-L1- versus 7 vehicle-treated mice. **f, g** msR4M-L1 treatment reduces lesions in aortic root. Representative images (**f**) and quantification (**g**) of ORO-stained sections from 12 msR4M-L1- versus 12 vehicle-treated mice. **h, i** msR4M-L1 treatment reduces macrophage content in aortic root. Representative images (**h**) and quantification of macrophage area (**i**) of anti-MAC2-stained (red) sections from 11 msR4M-L1- versus 12 vehicle-treated mice. **j** msR4M-L1 reduces circulating inflammatory cytokines. Cytokine-array analysis of plasma samples from 6 msR4M-L1- versus 6 vehicle-treated *Apoe*$^{-/-}$ mice on HFD (duplicates each) (means ± SD). **k–m** MIF gene expression is upregulated in atherosclerotic plaques from CEA patients compared to healthy vessels but no difference between plaque stages. Expression measured with mRNA from paraffin sections (**k**, qPCR, n = 19 stable and 20 unstable plaques, n = 4 healthy vessels) and mRNA from fresh tissue (**l**, qPCR, n = 9 early, n = 9 advanced plaques; **m**, RNAseq, n = 6 stable and n = 5 unstable plaques). **n–p** Side-by-side-comparison (representative images) between anti-MIF antibody-based IHC and TAMRA-msR4M-L1 staining for selected sections of stable CEA plaques (**n**, n = 6 sections, 11 specimens), unstable plaques (**o**, n = 2 sections, 15 specimens), and healthy vessel (**p**, n = 2 sections, 6 specimens); (left, overviews; right, magnifications of selected areas (boxes 1, 2) stained by anti-MIF antibody (brown) and TAMRA-msR4M-L1 probe (red), DAPI + (blue); circular arrow indicates DAB-/TAMRA-stained slides are not in same orientation). Quantification of TAMRA-msR4M-L1-stained full cohorts in Supplementary Fig. 27b. Scale bars: **d** 50 µm; **f** 200 µm; **h** 200 µm; **n–p** 100 µm. Data in **e, g, i**, and **j** are reported as means ± SD; data in **k–m** as box-whisker plots. Statistical analysis performed with unpaired two-tailed t-test (**e, g, i, j**), two-tailed Mann–Whitney test, or Kruskal–Willis test (**k, l, m**) as appropriate. msR4M-L1, MIF-specific CXCR4 mimic-L1; MIF, macrophage migration-inhibitory factor. Source data including definitions of box-plot parameters (minima, percentiles, centers, maxima) of **k–m** are provided as a Source Data file.

toward reduction in the msR4M-L1-treated group (Supplementary Table 3, Supplementary Fig. 24). Importantly, atherosclerotic lesion size in aortic arch (Fig. 6d, e) and root (Fig. 6f, g) was significantly decreased in msR4M-L1-treated mice compared with vehicle-treated controls. Moreover, protection from lesion formation was accompanied by a significantly decreased number of lesional macrophages in the msR4M-L1 group, as revealed by MAC-2 staining (Fig. 6 h, i), and by a marked reduction in circulating inflammatory cytokine levels, as measured by a cytokine array (Fig. 6j; Supplementary Fig. 25). Reductions in

the msR4M-L1-treated group were seen for IL-1α, IL-16, TNF-α (Fig. 6j) as well as CXCL13/BLC (Supplementary Fig. 25), with trends observed for IL-27, CCL12, IFN-γ, and TIMP-1, indicating that the CXCR4 mimic broadly down-regulates the inflammatory response associated with atherogenesis. Together, this demonstrated that msR4M-L1 exhibits a therapeutic atheroprotective and anti-inflammatory capacity in an experimental in vivo model of atherosclerosis.

To further test the translational relevance of these findings, we probed human carotid atherosclerotic plaque sections obtained

from patients undergoing carotid endarterectomy (CEA). Based on the histological characterization of plaque morphology and availability of specimens for mRNA analysis, stable and unstable plaques, and early versus advanced stage lesions were examined. Healthy vessels served as additional controls (Supplementary Table 4). MIF expression has been amply characterized in atherogenic mouse models, but there is only limited information on the quantification of MIF expression in human CEA lesions[45]. We first profiled MIF mRNA expression from both paraffin-embedded tissues and fresh plaque specimens by qPCR and RNA sequencing (RNAseq). Quantitative PCR with mRNA extracted from paraffin sections suggested that MIF expression was markedly upregulated in both stable and unstable CEA plaques, when compared to healthy control tissue (Fig. 6k). To further characterize MIF expression in CEA plaques, fresh human tissue specimens were studied by qPCR and RNAseq. These analyses confirmed that MIF expression did not significantly differ between early and advanced plaque stages (Fig. 6l, Supplementary Fig. 26) or between stable and unstable plaques (Fig. 6m), respectively. Selected specimens based on the mRNA expression data were next applied for a side-by-side alignment of MIF expression comparing immunohistochemical (IHC) staining of MIF protein with an established antibody and staining with TAMRA-msR4M-L1 as probe (Fig. 6n–p, Supplementary Figs. 27 and 28). Of note, staining with the CXCR4 mimic paralleled the detection of MIF expression by IHC, exhibiting pronounced staining of stable and unstable plaque tissue that was markedly higher than TAMRA-msR4M-L1 positivity in healthy vessels (Fig. 6n–p, respectively). Furthermore, quantification of TAMRA-msR4M-L1 signals from stable and unstable plaque specimens versus healthy vessel tissues was in line with the qPCR and RNAseq-based expression pattern (Supplementary Fig. 28, Fig. 6k–m). Moreover, MIF-specificity of the CXCR4 mimic-derived probe was verified by a competition experiment with peptide MIF(54–80), which is part of core binding region and competes with MIF/msR4M-L1 binding (see Fig. 2h), and significantly reduced stable plaque staining by TAMRA-msR4M-L1 (Supplementary Fig. 28). Interestingly, previous work had shown that CXCL12 exhibits more pronounced expression in unstable compared to stable plaque[46]. Together, these findings confirmed the significance of msR4M-L1/MIF binding in the context of atherosclerotic plaque tissue from human CEA specimens.

## Discussion

Anti-cytokine/-chemokine strategies represent promising therapeutic approaches for a variety of diseases, including cancer, inflammation, and cardiovascular diseases. In addition to SMDs and antibodies, soluble receptors are an important targeting approach to block pathogenic cytokine effects[7,47]. While soluble cytokine receptors have been developed for single-membrane-spanning receptors and are successfully used in the clinic against immune-mediated diseases, anti-chemokine strategies based on a soluble receptor principle are not established.

Here we report on a small engineered peptide-based, soluble chemokine receptor mimic that distinguishes between two chemokines and features ligand- and receptor-selective anti-atherosclerotic capacities in vitro and in vivo. We focused on CXCR4, one of the most studied chemokine receptors[48,49]. CXCR4 has critical ligand- and context-dependent roles in various diseases. Together with its ligand CXCL12, it is a promising target in tumor metastasis[48] and small molecule CXCR4 inhibitors such as Plerixafor/AMD3100 are used as stem cell mobilizers for transplantation therapy of patients with specific cancers[50]. However, in atherosclerotic diseases, the CXCR4/CXCL12 axis has proven to

be a difficult target, with both disease-promoting and protective properties. Genome-wide association studies (GWAS) and CXCL12 plasma level analysis revealed *CXCL12* as a candidate gene associated with CAD[51–53], and disease-exacerbating activities such as cardiac inflammatory cell recruitment have been implied for the CXCR4/CXCL12 axis[54,55]. In contrast, beneficial activities include cardioprotective effects based on the contribution of CXCR4/CXCL12 to neoangiogenesis and cardiomyocyte survival[49,56,57]. Moreover, disruption of this axis promotes atherosclerotic lesion formation through deranged neutrophil homeostasis[58] and loss of atheroprotection[26]. In this context, we have shown that atherogenesis-induced endothelial damage is counter-acted by unleashing CXCR4 activity and autocrine CXCL12 expression in endothelial cells through miR-126-containing apoptotic bodies[27] and that CXCR4 on vascular cells maintains arterial integrity and limits atherosclerosis by preserving barrier function and a normal contractile vascular smooth muscle cell (VSMC) phenotype[26].

Capitalizing on our earlier findings that CXCR4 engages MIF as a non-cognate ligand to drive atherogenic leukocyte recruitment[14,16,17] and that CXCR4-supported endothelial barrier integrity is mediated by CXCL12 but not MIF[26], we surmised that MIF-specific CXCR4 targeting might be a promising avenue to circumvent the complexity of the CXCR4/CXCL12 system in cardiovascular conditions. In fact, we previously demonstrated that MIF-blocking strategies are superior to CXCL12 blockade in inducing plaque regression[6,14] and that a native LDL/LDL receptor (LDLR)-based foam cell-promoting activity of CXCR4 is primarily elicited by MIF and not CXCL12[41]. However, currently available MIF-blocking strategies may not be optimal, as anti-MIF (Imalumab) or anti-CD74 (Milatuzumab) antibodies would potentially interfere with the cardioprotective MIF/CD74 axis[15,16]. The same holds true for MIF-directed SMDs, which are designed to bind in MIF's conserved tautomerase pocket and interfere with MIF binding to CD74. Indeed, modification of this cavity invokes conformational changes in MIF that impair binding to CD74[59]. AMD3100 partially interferes with MIF/CXCR4 binding[14,25], but this CXCR4 inhibitor has been found to impair the cardio- and atheroprotective activity spectrum of the CXCR4/CXCL12 axis[26,27,58].

Our design was guided by the CXCR4 structures[29–31] and SAR studies[24,25], highlighting CXCR4 ectodomain regions that may be harnessed to engineer a soluble receptor mimic to selectively target MIF and spare CXCL12. Approaches to utilize the ectodomains of single-membrane-spanning type I cytokine receptors such as the TNF or IL-6 receptor have been successfully developed as immunomodulatory drugs[7,47]. However, mimicking the ectodomain of seven-helix membrane-spanning GPCRs is inherently complex due to the discontinuous nature of the receptor backbone topology. Ligand binding in (poly)peptide-ligating GPCRs such as chemokine receptors typically involves several extracellular portions of the receptor, often a combination of residues of several ECLs and the N-terminal[30]. Only a handful of reports are available: the N-terminal and ECL3 elements of CXCR1 and CXCR2 were assembled on a soluble GPCR B1 domain scaffold protein[60]; based on the crystal structure of rhodopsin, all three predicted ECLs of CXCR4 were connected to form an HIV gp120-binding mimic[61]; and a construct mimicking corticotropin-releasing factor receptor-1 (CRF-R1) combined native chemical ligation and recombinant technology and encompassed the entire 23 kDa ectodomain of CRF-R1[62]. Such studies have remained explorative, led to constructs with micromolar binding affinities, and neither chemokine selectivity nor in vivo or disease relevance were addressed.

The engineered MIF-selective CXCR4 mimics reported here are highly reduced GPCR mimics of only 29 residues plus two

non-natural amino acids of the linker moiety (molecular weight <4 kDa), reducing the size of CXCR4 by >90%. MIF selectivity over CXCL12 was achieved by combining only selected residues within ECL1 and ECL2. As determined by independent biophysical methods such as fluorescence spectroscopy, FP, and MST, the lead candidate mimics bind MIF with low nanomolar affinity ($K_D$ ~ 20–75 nM), in line with the reported $K_D$ value of 19 nM for MIF binding to full-length membrane CXCR4[14], while binding to CXCL12 is essentially absent. This affinity is reasonable compared with that of Imalumab or the pre-clinical anti-MIF antibody NIH/III.D9 ($K_D$ ~ 1–3 nM)[63] and MIF-directed SMDs (micromolar $K_D$)[16]. Of note, msR4M-L1 neither affected MIF binding to CD74 as indicated by FP and MST, nor did it impair MIF/CD74-mediated stimulation of AMPK phosphorylation in human cardiomyocytes as a correlate of MIF's cardioprotective activity[15]. Hence, msR4M-L1 has more favorable selectivity characteristics than the available anti-MIF therapeutic strategies. A molecular explanation for this selectivity comes from experiments mapping the MIF binding site by MIF peptide fragments and a site-specific mutant. msR4M-L1 targets MIF region 54–80, a part of the N-like loop known to mediate MIF/CXCR4 binding, but not involved in MIF/CD74 binding, in line with data showing that the tautomerase site of MIF, Tyr-100, and residues 80–87 determine the MIF/CD74 binding interface[21,22,59]. Importantly, binding selectivity of msR4M-L1 for MIF versus CXCL12 was functionally paralleled in a number of inflammation- and atherosclerosis-relevant cell systems, i.e. GPCR/CXCR4 signaling, 2D lymphocyte chemotaxis, foam cell formation, monocyte adhesion under static and flow conditions, and 3D monocyte migration, together representing MIF/CXCR-mediated cell systems with disease relevance[41].

Mapping of the binding site of the CXCR4 mimic by alanine-scanning pinpointed a role for the aromatic residues within msR4M-L1, in particular Phe-104 and Phe-107, and also offered additional proof for the specificity of the MIF/msR4M-L1 binding mechanism. Interesting structural information also comes from mimics, in which we introduced a disulfide bridge between residues Cys-109 of ECL1 and Cys-186 of ECL2. In contrast to msR4M-L1 and -L2 that are fully selective for MIF, introduction of the disulfide bridge led to a gain-of-CXCL12-binding activity, irrespective of the presence (msR4M-L1ox, msR4M-L2ox) or absence (msR4M-LS) of the spacer-mediated conformational constraint. This is in line with the previously identified Cys-109-Cys-186 disulfide in the X-ray structure of CXCR4[29,30] and existing knowledge on the CXCR4/CXCL12 interface[64], and supports the notion that the natural CXCR4 receptor is 'equipped' to interact with both CXCL12 and MIF[14]. On the other hand, the $K_D$ for MIF binding dropped >10-fold, when the respective ECL1 and ECL2 sequences were not covalently linked. Together, these data reveal that while the herein identified aromatic residues and the conformational restriction imposed on ECL1 and ECL2 are required for MIF binding, a certain degree of conformational flexibility is necessary to guarantee selectivity between different CXCR4 chemokines. Comparison of the various synthesized mimics further instructs for future optimization towards higher potency, stability, or selectivity[28].

The biochemical and cell-based experiments encouraged us to examine whether the mimics would be efficacious in a pathogenic ex vivo organ or in vivo setting. Using fluorescently labeled msR4M-L1 to stain atherosclerotic tissue sections from atherogenic mice and in vivo administration of labeled peptide verified that msR4M-L1 localizes to atherosclerotic plaque tissue in a MIF-specific manner. Indeed, MIF has been shown to be up-regulated in atherosclerotic lesions, where secreted MIF is deposited similar to classical arrest chemokines and localizes to plaque macrophages, foam cells, and VSMCs[14,45,65]. While these experiments do not fully exclude the possibility that msR4M-L1 also -partially- localizes to CXCL12 + regions, our biochemical data proving binding selectivity, suggest that this is unlikely. Furthermore, while the MIF homolog MIF-2/D-DT[13] has not been studied in atherosclerosis, it may be of future interest to design mimics directed at MIF-2 for applications in MIF-2-dominated inflammatory conditions.

We used an MPM-based ex vivo atherosclerotic carotid artery system to monitor luminal leukocyte adhesion under pathophysiologically relevant conditions and demonstrated that treatment with msR4M-L1 markedly attenuated adhering leukocyte numbers. Such systems have been powerful in demonstrating the leukocyte recruitment potential of MIF or classical arrest chemokines such CXCL1/KC[14,44,58,66,67]. In conjunction with the Fluos-msR4M-L1 plaque staining data, the MPM data indicate that msR4M-L1 blocks MIF-mediated atherogenic leukocyte recruitment. Important proof for a translational utility of the GPCR mimics reported here comes from testing msR4M-L1 therapeutically in a mouse model of atherosclerosis in vivo[44]. The chosen treatment regimen of three 50 μg-injections per week was instructed by plasma proteolytic and in vivo stability data that indicated reasonable plasma levels of msR4M-L1 in the range of 20–50 nM up to two days after injection, maintaining circulating doses of the mimic overall in line with the determined $K_D$/IC$_{50}$ values. The mimic potently blocked atherosclerosis at key pre-dilection sites, reduced lesional macrophage accumulation and circulating inflammatory cytokines/chemokines, while no effects on lipids or leukocyte counts were observed, suggesting that it specifically targeted a MIF-mediated pathogenic inflammatory effect in atherogenic lesions. The experiment constitutes a 'proof-of-concept' for such compounds in an in vivo disease setting and is a good predictor for their efficacy in advanced atherosclerosis models, but also other models involving MIF-related chronic inflammation[12,14,16,17,58,68]. Moreover, CXCR4 is a major receptor driving cancer metastasis, and not only the CXCR4/CXCL12 but also the CXCR4/MIF axis has been implicated in this process[48,69]. While it is beyond the scope of this study to address the inhibitory potential of our peptides in cancer, the mimics appear principally suitable for such an application. Further, the selectivity differences seen between our covalently linked conformationally restricted versus non-linked versus hyper-restricted constructs may instruct for the design of dual-specificity inhibitors against both MIF and CXCL12, e.g. for future applications in cancer. Similarly, it may be envisaged to expand the concept to MIF/CXCR2, which also has a role in atherosclerosis[1,14].

The CANTOS trial has provided clinical proof that an immunotherapy-based targeting approach against IL-1β, a key inflammatory mediator, improves cardiovascular outcome in an at-risk population[4,70]. However, treatment with Canakinumab did not improve mortality in atherosclerotic patients and caused an increase in infections, highlighting the need to identify additional drug targets and to develop anti-inflammatory strategies with a high selectivity profile that block atherosclerotic pathways. Engineering of CXCR4 mimics specifically targeting MIF/CXCR4 interactions could be one such approach and represent a promising class of anti-atherogenic molecules based on the applied soluble GPCR ectodomain concept. MsR4Ms are peptide-based molecules and, while there are over 60 peptide drugs approved worldwide, there are pros and cons compared to antibodies and SMDs. Advantages are good surface coverage and hence high selectivity and potency, favorable safety, and low-cost production; disadvantages are the limited proteolytic stability and bioavailability[71]. However, these issues can be overcome by peptide chemistry tools and peptide design strategies[28,71]. Thus, msR4M-L1 should be viewed as a proof-of-concept inhibitor of MIF/CXCR4-specific atherogenesis, whose properties may be

improved by designed second-generation mimics. Of note, despite the many inherent differences between the experimental atherosclerosis mouse model applied herein and future clinical applications and the numerous unforeseeable risks along any drug development pipeline, the in vivo msR4M-L1 dose and application frequency used in our study overall appears to be compatible with a potential optimization route towards an efficacious drug. Accordingly, studies in patients with atherosclerotic disease could be a future perspective. In fact, TAMRA-msR4M-L1 staining of human carotid artery samples from patients who underwent CEA mirrored the high MIF expression profile in both stable and unstable plaque lesions versus healthy vessels as detected by qPCR, RNAseq, and antibody staining, but differs from the expression profile of CXCL12 as previously determined by qPCR[46].

In conclusion, the designed MIF-selective soluble CXCR4 mimics are a previously unrecognized class of anti-atherosclerotic/-inflammatory agents that could complement currently available inhibition strategies by antibodies or SMDs. We demonstrate that these molecules can be engineered to be chemokine-selective, to exhibit high binding affinities, and to be potent in blocking atherogenic chemokine activities in vitro and in vivo, while sparing potentially contraindicative protective pathways through alternative receptors or ligands.

## Methods

**Cytokines/chemokines and reagents**. Biologically active recombinant MIF was prepared by expression in the pET11b/E. coli BL21/DE3 system, recovery of the supernatant of the bacterial lysate, centrifugation, filtration, purification by Mono Q anion exchange and C8 reverse-phase chromatography, and dialysis-based renaturation and exhibited a purity of ~98%. The details of this procedure are described in references[14,36]. For some of the biophysical methods, a 90–95% purified preparation was used. Fluorescently labeled MIF[18] (Alexa-488-MIF or Alexa-647-MIF) was generated using Microscale Protein Labeling Kits from Invitrogen-Molecular Probes (Karlsruhe, Germany; Alexa-488/A30009, Alexa-647/A30009). LPS content was tested by limulus amoebocyte assay (LAL, Lonza, Cologne, Germany) and verified to be <5 pg μg$^{-1}$. Mutant [A$^{47-56}$]-MIF or MIF (10xAla) was generated by PCR cloning using the cDNA sequence of wildtype human MIF (WT-MIF) as template and the sequence verified by DNA sequencing. Bacterial expression and purification of MIF(10xAla) were performed following the established protocol for WT-MIF[36], except that pET29b was used and the bacterial-expressed protein initially recovered from inclusion bodies. Cell culture-grade tumor necrosis factor (TNF)-α was purchased from Life Technologies (Carlsbad, United States). Recombinant CXCL12 was a gift of Dr. von Hundelshausen (LMU Munich)[67] or was purchased from Peprotech (Hamburg, Germany). Other reagents were obtained from Sigma, Merck, Roth, or Calbiochem, and were of the highest purity degree available.

**Peptide design, peptide synthesis, purification, and linker chemistry**. Based on the crystal structures of human CXCR4 (accession codes 3ODU; 3OE0; 3OE6; 3OE8; 3OE9; 4RWS and previous SAR studies[24,25], CXCR4 ectodomain peptides were selected. The crystal structures were imported into PyMOL Molecular Graphics System (Version 1.8.2.2 Schrödinger, LLC) and Jmol (http://www.jmol.org) for determining the C-to-N distance between residues 97–110 and 182–196[29,30]. Conjugates of 12-Ado with either 6-Ahx or O2Oc were visualized in three-dimensional space using Molview and Jmol. The estimated distances between the N- and C-terminal in both conjugates were similar to the ECL1-ECL2 distance. All CXCR4-derived peptides, including msR4M-L1(7xAla) and msR4M-L1(2xAla), were synthesized as C-terminal amides on Rink amide MBHA resin by SPPS using Fmoc chemistry[28]. Couplings of Fmoc-6-Ahx-OH, Fmoc-12-Ado-OH and Fmoc-O2Oc-OH (Iris Biotech GmbH, Marktredwitz, Germany) were carried out with 3-fold molar excess of 2-(7-Aza-1H-benzotriazole-1-yl)- 1,1,3,3-tetramethyluronium hexafluorophosphate (HATU) and 4.5-fold molar excess of N,N-diisopropylethylamine (DIEA) in N,N-dimethylformamide (DMF). Fmoc-deprotection was in general carried out with 0.1 M hydroxybenzotriazole (HOBt) in 20% v/v piperidine in dimethylformamide (DMF) for 3 and 9 min to avoid aspartimide formation[32]. To introduce N$^\alpha$-fluorescein and TAMRA labels, 5(6)-carboxyfluorescein-N-hydroxysuccinimide ester (Sigma-Aldrich, Taufkirchen, Germany) and 5(6)-carboxytetramethylrhodamine (TAMRA, Novabiochem/Merck KGaA, Darmstadt, Germany) were coupled N-terminally to side chain-protected msR4M-L1 on solid phase, after Fmoc-deprotection[32]. The N$^\alpha$-biotinyl label was introduced as follows: after assembly of fully protected msR4M-L1 and N$^\alpha$-Fmoc-cleavage, Fmoc-protected 6-Ahx was coupled followed by Fmoc-cleavage and coupling of biotin using the coupling protocols as above[32]. Disulfide bridges in msR4M-L1ox and

msR4M-L2ox were formed in 1 mg mL$^{-1}$ peptide solution in aqueous 3 M guanidinium hydrochloride (GdnHCl) and 0.1 M ammonium carbonate (NH$_4$HCO$_3$) solution, containing 40% dimethylsulfoxide (DMSO). msR4M-LS was produced similarly, using 0.3 mg mL$^{-1}$ ECL1 and 0.5 mg mL$^{-1}$ ECL2 and 20% DMSO. Reverse-phase high-performance liquid chromatography (RP-HPLC) was applied for the purification of crude and oxidized peptides by using Reprosil Gold 200 C18 (250×8 mm) or Reprospher 100 C18-DE (250 × 8 mm) columns with pre-column (30×8 mm) (Dr. Maisch-GmbH, Herrenberg, Germany). The mobile phase consisted of 0.058% (v/v) trifluoroacetic acid (TFA) in water (buffer A) and 0.05% (v/v) trifluoroacetic acid in 90% (v/v) acetonitrile and water (buffer B) (flow rate 2.0 mL/min). All peptides were purified with an elution program of 10% B for 1 min, followed by a gradient from 10 to 90% B over 30 min, except for msR4M-LS, which was eluted with 30% B for 7 min followed by an increase to 60% B over 30 min. Expected molecular weights were verified by matrix-assisted laser desorption/ionization mass spectrometry (MALDI-MS). Peptides were used as TFA salts. For in vivo experiments, the TFA anion was exchanged to chloride by four cycles of dissolution/lyophilization of pure msR4M-L1 in aqueous 5 mM HCl and one cycle of bidistilled water. MIF sequence-based peptides (Supplementary Table 2) were synthesized on Wang resin or purchased from Peptide Specialities GmbH (PSL, Heidelberg, Germany). MIF-derived peptides were N-terminally acetylated and their C-terminal end is a free carboxylate.

**Fluorescence spectroscopy**. Fluorescence spectroscopic titrations were performed using a JASCO FP-6500 fluorescence spectrophotometer. MIF, MIF(10xAla), or CXCL12 were reconstituted in 20 mM sodium phosphate buffer, pH 7.2; peptide stocks including those of MIF fragments were freshly made in 1,1,1,3,3,3-hexafluoro-2-propanol (HFIP) at 4 °C at a concentration of 1 or 2 mM[28,32]. After adding Fluos-labeled peptides or Alexa-488-MIF and their unlabeled titration partner in assay buffer and a subsequent mixing step, measurements were performed in 10 mM sodium phosphate buffer, pH 7.4, containing 1% HFIP. Fluos-labeled peptide was applied at 5 nM and Alexa-488-MIF at 10 nM unless indicated otherwise. For the titration with ISO-1, Alexa-488-MIF had a concentration of 50 nM and ISO-1 was varied from 0.1 to 500 μM in 10 mM sodium phosphate buffer, pH 7.4, containing 0.5% DMSO. The excitation wavelength was 492 nm and emission spectra were obtained between 500 and 600 nm. Apparent $K_D$ values (app. $K_D$) were calculated assuming a 1/1 binding model using sigmoidal curve fittings with OriginPro 2016 (OriginLab, Northampton, MA, USA) or GraFit 5 (Erithacus Software Ltd., Wilmington House, UK) data analysis software as appropriate[32].

**Circular dichroism (CD) spectroscopy**. CD spectra were obtained with a JASCO J-715 spectropolarimeter (JASCO, Tokyo, Japan) applying an established protocol[72]. Briefly, far-UV CD measurements were carried out between 195 and 250 nm. The response time was set at 1 s, intervals at 0.1 nm, and bandwidth at 1 nm. All spectra were measured at RT and represent an average of three recorded spectra. Scans were recorded for the ectodomain mimic peptides at a concentration of 1–20 μM in 10 mM sodium phosphate buffer, pH 7.4, containing 1% HFIP, following dilution of freshly made peptide stock solution in HFIP (4 °C) into the buffer-containing cuvette. Singular ECL1 and ECL2 were measured at 5 μM. The background spectrum of buffer/1% HFIP alone was subtracted from the spectra of the peptides. Dynode voltage was below 1000 and did not interfere with the measurements.

**Fluorescence polarization**. FP was measured with a JASCO FP-6500 fluorescence spectrophotometer equipped with FDP-223 and FDP-243 polarizers. Stock solutions for peptides, MIF, and CXCL12 were prepared as described above under fluorescence spectroscopy. Final mixtures between Fluos-msR4M-L1 (5 nM) and non-labeled MIF (titrated from 0.5 to 277.5 nM) or CXCL12 (varied from 5 nM to 2.5 μM) were measured in 10 mM sodium phosphate, pH 7.2, containing 0.5% HFIP. Alexa-488-MIF (10 nM) and unlabeled peptide (titrated from 0.5 nM to 1 μM) was measured in 10 mM sodium phosphate, pH 7.2, containing 2% HFIP. Experimental conditions were similar for the titrations between Alexa-488-MIF and soluble human CD74 (sCD74). Soluble CD74[19] is a fusion protein of an N-terminal HA-tag and CD74 residues 73–232 (R&D Systems, Minnesota, USA). The sCD74 stock solution (26 μM) was prepared in PBS, pH 7.2. Alexa-488-MIF was prepared in 20 mM sodium phosphate buffer, pH 7.2. For binding analyses, 10 nM Alexa-488-MIF and the respective sCD74 sub-stocks were mixed in 10 mM sodium phosphate, 2% HFIP, 0.01 × PBS, and incubated at RT for 4 h. For competition experiments with msR4M-L1, Alexa-488-MIF was briefly pre-incubated with a 20-fold molar excess of msR4M-L1, and the mixture titrated against sCD74 as above. Bandwidth for excitation and emission was set at 5 nm and time response at 0.5 s. The excitation wavelength was 492 nm and emission was recorded at 519 nm. Measurements were performed at RT and within 2–3 min upon preparation of the solutions. Polarization was calculated using the equation: $P = (I_\parallel − G \cdot I_\perp) / (I_\parallel + G \cdot I_\perp)$, in which $I_\parallel$ is the intensity of emitted light polarized parallel to the excitation light, $I_\perp$ is the intensity of emitted light polarized perpendicular to the excitation light, and the G factor was calculated based on the instrumental documentation (https://currentprotocols.onlinelibrary.wiley.com/doi/10.1002/9780470559277. ch090102. Apparent $K_D$ values (app. $K_D$) were calculated assuming a 1/1 binding

model, using sigmoidal curve fittings with OriginPro 2016 (OriginLab) or GraFit 5 (Erithacus Software Ltd.) data analysis software as appropriate[32].

**Dot blot**. Different amounts (0–400 ng) of human MIF, mouse MIF, or human CXCL12 were spotted on a nitrocellulose membrane and membranes allowed to dry for 30 min. Non-specific binding was blocked with Tris-buffered saline (TBS), pH 7.4, containing 0.1% Tween-20 (TBS-T) and 1% BSA. TAMRA-msR4M-L1 was reconstituted at a concentration of 10 μM in PBS containing 2.5% HFIP, diluted to a 3 μM working solution in 1% BSA/TBS-T, and incubated with the membrane at 4 °C. For the competition experiment, TAMRA-msR4M-L1 was incubated in the presence of a 2-fold molar excess of MIF peptide fragment MIF[54–80]. Fluorescence intensities were measured at 600 nm using an Odyssey® Fc imager (LICOR Biosciences, Bad Homburg, Germany). The total intensity of each spot was automatically corrected by the individual background signal. The signal intensity of 400 ng human MIF was set to 100%.

**Microscale thermophoresis**. MST measurements were recorded on a Monolith NT.115 instrument with green/red filters (NanoTemper Technologies, Munich, Germany). MST and LED power were set 80 and 95%, respectively. All measurements were performed at 37 °C. MST traces were tracked for 40 s (laser-off: 5 s, laser-on: 30 s; laser-off: 5 s). A stock solution of 200 nM TAMRA-msR4M-L1 was prepared in 20 mM sodium phosphate buffer, pH 7.2, containing 0.2% Tween-20. For titration of MIF, sub-stock solutions were prepared by serial 1:1 dilutions from a 20 μM stock solution in 20 mM sodium phosphate buffer, pH 7.2. TAMRA-msR4M-L1 (final concentration: 100 nM) and each MIF sub-stock were mixed at a 1:1 ratio, incubated for 10 min and loaded in the capillaries. Data points/MST traces were analyzed at an MST-on time of 1.5 s using the MO.Affinity Analysis v2.2.4 (NanoTemper Technologies). The same conditions were used for analyzing titrations between TAMRA-msR4M-L1 (100 nM) and CXCL12.

Experimental conditions were similar for the titrations between Alexa-647-MIF and soluble human CD74 (sCD74), except that LED power was set at 80%. The sCD74 stock solution (26 μM) was prepared in PBS, pH 7.2, and then further diluted in 10 mM Tris and 0.5 × PBS, containing 0.01% BSA. The Alexa-647-MIF stock was 40 nM in 10 mM Tris buffer, pH 8.0. For binding analyses, Alexa-647-MIF was mixed 1:1 with the respective sCD74 sub-stocks and mixtures incubated at RT for 3 h and 37 °C for 15 min prior to analysis at the same temperature. Data analysis was based on the signal of an MST-on time of 30 s. For competition experiments with msR4M-L1, 40 nM Alexa-647-MIF was pre-incubated with 400 nM msR4M-L1 for 10 min, and the mixture titrated against sCD74 as above.

App. K$_D$ values were calculated assuming a 1/1 binding model, using sigmoidal curve fittings with OriginPro 2016 (OriginLab) or GraFit 5 (Erithacus Software Ltd.) data analysis software as appropriate[32].

**Peptide spot array technology and alanine-scanning**. Peptides were generated by stepwise SPOT synthesis on modified cellulose disks (Intavis MultiPep RSi/ CelluSpot Array, Cologne, Germany) essentially as described previously[24]. Briefly, peptides were processed, cellulose dissolved, and spotted onto coated glass slides using an Intavis Slide Spotting Robot. Glass microarrays were blocked in 50 mM Tris-buffered saline (TBS) containing 0.1% Tween 20 and 1% BSA and washed in TBS containing 0.1% Tween 20. The array was probed with 3 μM biotinylated human MIF and developed with horse-radish peroxidase (HRP)-conjugated streptavidin in blocking buffer. For the determination of false-positives, a control array was incubated with HRP-conjugated streptavidin alone. Chemiluminescent signals were measured by Odyssey® Fc imager. The intensity of each spot was corrected for spot-specific background signal and normalized to the intensity of the wildtype peptide.

**CXCR4-specific signaling in a yeast-based cell system**. The yeast CXCR4-specific cell signaling system employing *S. cerevisiae* strain (CY12946), expressing functional CXCR4 that replaces the yeast STE2 receptor and is linked to a β-galactosidase (lacZ) signaling read-out, was used, as CXCL12 and MIF elicit a CXCR4-specific signaling response in this cell system[24,25]. Briefly, yeast transformants stably expressing human CXCR4 were grown overnight at 30 °C in yeast nitrogen base selective medium (Formedium, UK). Cells were diluted to an OD$_{600}$ of 0.2 and grown to an OD$_{600}$ of 0.3–0.6. Transformants were incubated with 20 μM human MIF or 2 μM human CXCL12 in the presence or absence of different concentrations of msR4M-L1 for 1.5 h. OD$_{600}$ was measured and activation of CXCR4 signaling quantified by β-galactosidase activity using a commercial BetaGlo Kit (Promega, Mannheim, Germany).

**Cell culture and cell lines**. Human aortic endothelial cells (HAoECs) were from PromoCell (Heidelberg, Germany). Cells were plated on collagen (Biochrom AG, Berlin, Germany) in endothelial cell growth medium (ECGM, PromoCell) and cultured according to the manufacturer's recommendations and as described previously[73]. The monocytic cell line MonoMac-6 was cultured in RPMI 1640 medium with 10% fetal calf serum (FCS). Primary human cardiac myocytes (HCM) isolated from the ventricles of the adult heart were from PromoCell and used at passage 2–8. They were cultured in myocyte basal medium (PromoCell), containing 5 μg mL$^{-1}$ insulin, 5% FCS, 2 ng mL$^{-1}$ fibroblast growth factor (FGF),

and 0.5 ng mL$^{-1}$ epidermal growth factor (EGF). Human embryonic kidney (HEK)-293 cells were cultured in DMEM-GlutaMAX (Life Technologies-Gibco) supplemented with 10% FCS and 1% penicillin/streptomycin. FCS was obtained from Invitrogen-Thermo Fisher Scientific. Miscellaneous cell culture reagents (media, supplements) were bought from Invitrogen and PAA (Pasching, Austria).

**HEK293-CD74 surface binding assay**. HEK293 cells were transiently transfected with 8 μg of the pcDNA3.1-CD74minRTS-FLAG plasmid using Polyfect (Qiagen, Hilden, Germany) and expressed surface CD74 after 24 h (efficiency 50–60%) in line with previous data[74]. HEK293-CD74 transfectants were washed and 3 × 10$^5$ cells resuspended in ice-cold PBS containing 0.5% BSA, and incubated with 400 nM Alexa-488-labeled MIF in the presence or absence of msR4M-L1 (2 μM) on ice for 2 h. After washing in ice-cold PBS containing 0.1% BSA, the amount of Alexa-488-labeled MIF bound to the cell surface was quantified by flow cytometry using a FACS Verse instrument (BD Biosciences, Heidelberg, Germany). Binding of Alexa-488-MIF to non-transfected wildtype HEK293 cells, which do not express CD74, served as background control.

**Mice**. Mice were housed under standardized light-dark cycles in a temperature-controlled air-conditioned environment under specific pathogen-free conditions at the Center for Stroke and Dementia Research (CSD), Munich, Germany, with free access to food and water. All mice used in this study were between 7 and 10 weeks of age and were on C57BL/6 background. *Apoe*$^{-/-}$ mice were initially obtained from Charles River Laboratories (Sulzfeld, Germany) and backcrossed within the CSD animal facility before use. The atherogenic *Ldlr*$^{-/-}$ and *Ldlr*$^{-/-}$ *Mif*$^{-/-}$ mice as well as *Apoe*$^{-/-}$ *Mif*$^{-/-}$ mice have been described previously[14,66]. All mouse experiments were approved by the Animal Care and Use Committee of the local authorities (Regierung von Oberbayern, ROB; Aktenzeichen Az = ROB-55.2Vet-2532.Vet_02–18–40) and performed in accord with the animal protection representative at the Center for Stroke and Dementia Research (CSD).

**Chemotaxis analysis of murine B cells**. A Transwell-based assay was used as described previously[35]. Briefly, splenic B cells were isolated by negative depletion using a Pan B Cell Isolation Kit (Miltenyi Biotec, Bergisch Gladbach, Germany). Purity of the cells was between 95 and 99%. One-hundred μL of cell suspension containing 1 × 10$^6$ cells in RPMI 1640/5% FCS was loaded into the upper chamber of a Transwell insert. Filters were transferred into the lower chambers containing MIF or CXCL12 in the presence or absence of ectodomain peptides. Chemotaxis was followed for 4 h at 37 °C in a humidified atmosphere of 5% CO$_2$. Migrated cells were counted by flow cytometry using CountBright™ Absolute Counting Beads (Molecular Probes-Invitrogen).

**CD74 signaling in human cardiomyocytes**. HCMs (2 × 10$^5$ per well) were plated in 12-well plates and maintained for 2 days with cultured medium containing 5% FCS. Before stimulation, medium was replaced by fresh myocyte basal medium containing 0.05% FCS and HCMs rested for 16 h. Surface CD74 expression on HCMs was verified by flow cytometry (FITC-conjugated anti-human CD74 (1:100 dilution), FITC-IgG2 (isotype control) (BD Pharmingen), 1 h/4 °C in the dark, BD FACSVerse™ flow cytometer, FlowJo software). AMPK signaling was elicited by addition of 16 nM of human MIF and incubation at 37 °C for 60 min. To test for an influence of msR4M-L1, MIF was pre-incubated with 16 or 80 nM msR4M-L1 and mixtures added to HCMs. After treatment, cells were lysed and subjected to SDS-PAGE (10%)/Western blotting using NuPAGE® lithium dodecyl sulfate (LDS)/ dithiothreitol (DTT) lysis buffer-containing PhosSTOP™ reagent (Roche Applied Science). AMPK activation was revealed with an antibody against phosphorylated AMPK (anti-pAMPKα, 1:1000; Cell Signaling Technologies, Heidelberg, Germany) and total AMPKα (anti-AMPKα, 1:1000), as well as actin detected for standardization. Anti-rabbit horse-radish peroxidase (HRP)-conjugated antibody (1:10000, GE Healthcare, Freiburg, Germany) was used for development and signals quantitated by chemiluminescence using an Odyssey® Fc imager. Unprocessed scans of the Western blots are supplied in the Source Data file.

**Isolation of human peripheral blood-derived monocytes**. Human peripheral blood-derived monocytes were isolated following an established procedure[14]. Briefly, blood was collected from healthy donors or buffy coat obtained from the blood bank of LMU University Hospital, mixed 1:1 with PBS, and PBMCs isolated by Ficoll-Paque Plus gradient (GE Healthcare). Monocytes were purified by negative depletion using the Monocyte Isolation Kit II (Miltenyi). Monocyte purity was verified by flow cytometry using an anti-CD14 antibody (Miltenyi) and was 95–98%. Purified cells were suspended in RPMI 1640 medium supplemented with 10% FCS, 1% penicillin/streptomycin, 2 mM L-glutamine and 1% NEAA. The isolation of PBMCs from donor blood was approved by the local ethics committee of LMU Munich.

**3D migration of human peripheral blood-derived monocytes by time-lapse microscopy**. The 3D-migration behavior of human monocytes was assessed by time-lapse microscopy and individual cell tracking using the 3D chemotaxis μ-Slide system from Ibidi GmbH (Munich, Germany), adapting the established Ibidi

dendritic cell protocol for human monocytes. Briefly, isolated monocytes ($4 \times 10^6$ cells) were seeded in rat tail collagen type-I gel in DMEM and subjected to a gradient of MIF or CXCL12 (64 nM) in the presence or absence of msR4M-L1. Cell motility was monitored performing time-lapse imaging every 1 min at 37 °C for 2 h using a Leica inverted DMi8-Life Cell Imaging System equipped with a DMC2900 Digital Microscope Camera with CMOS sensor and live cell-imaging software (Leica Microsystems, Wetzlar, Germany). Images were imported as stacks to ImageJ software and analyzed with the manual tracking and chemotaxis/migration tools (Ibidi GmbH).

**DiI-oxLDL/DiI-LDL uptake/foam cell formation**. MIF/CXCR4-dependent foam cell formation was assessed by measuring uptake of fluorescently labeled oxidized or native human low-density lipoprotein particles (DiI-oxLDL or DiI-LDL, respectively) in primary human monocyte-derived macrophages essentially following an established protocol[41]. Briefly, cells were incubated in culture medium (RPMI 1640-GlutaMAx medium containing 100 U mL$^{-1}$ penicillin, 100 µg mL$^{-1}$ streptomycin, and 0.2% BSA) for 15 h at 37 °C and subsequently incubated in the same medium supplemented with 1% HPCD ((2-hydroxy)-β-cyclodextrin, Sigma-Aldrich) for 45 min. After washing with imaging solution (MEM without phenol red containing 30 mM HEPES, 0.5 g/L NaHCO$_3$, pH 7.4, and 0.2% BSA), cells were exposed to 50 µg mL$^{-1}$ 1,1'-dioctadecyl-3,3,3'3'-tetramethylindocarbocyanine-labeled oxidized LDL (DiI-oxLDL) or DiI-labeled native LDL (DiI-LDL) for 30 min at 4 °C, followed by incubation at 37 °C for 20 min. Cells were washed with ice-cold imaging solution (pH 3.5), fixed, and counter-stained with Hoechst 33258.

**Static monocyte adhesion**. HAoECs were seeded at a density of 30,000 cells/well in 6 well µ-Ibidi Perfusion slides VI 0.4 (Ibidi GmbH). After confluency was reached, TNF-α or MIF were added at a final concentration of 4 or 16 nM, respectively, in the presence *versus* absence of msR4M-L1 (320 nM), and cells incubated for 16 h. To mimic chronic atherogenic inflammatory conditions, cells were pre-treated with 40 pM human TNF-α before adding MIF. After perfusion of the chambers with fresh medium, MonoMac6 cells ($1 \times 10^6$ cells mL$^{-1}$) in PromoCell medium were added for 30 min. Non-adhering cells were flushed away by gentle perfusion using a 30 mL syringe. To quantify adherent monocytes, seven individual images (technical triplicates each) from each treatment were acquired using a Leica DMi8 inverted microscope with a ×10 objective and cells quantified using ImageJ.

**Monocyte adhesion under flow**. HAoECs were seeded at a density of 60,000 cells per channel in collagen-coated Ibidi µ-Slides I 0.8 and incubated for 20–24 h until monolayers were confluent. HAoECs were treated with 16 nM human MIF in the presence *versus* absence of msR4M-L1 (320 nM) for 2 h. MonoMac6 cells were exposed to human MIF and msR4M-L1 at the same concentrations for 2 min, before they were transferred into assay buffer (1 x HBSS, 10 mM HEPES, 0.5% BSA) at a density of $0.5 \times 10^6$ cells mL$^{-1}$ and kept at 37 °C. Prior to the flow assay, MgSO$_4$ and CaCl$_2$ were added to MonoMac6 suspensions at a final concentration of 1 mM. Flow channels containing HAoEC monolayers were then perfused with MonoMac6 cells at a shear rate of 1.5 dyn cm$^{-2}$ for 10 min at 37 °C using the Ibidi Pump System/Perfusion Set. For quantification of adherent monocytes, images from four positions per channel were acquired using a Leica DMi8 inverted microscope (20 x objective) and quantified with ImageJ.

**Staining of atherosclerotic plaque tissue with Fluos-msR4M-L1**. Immuno-fluorescent staining of atherosclerotic tissue with Fluos-msR4M-L1 was performed with specimens from atherogenic *Ldlr$^{-/-}$* and *Apoe$^{-/-}$* mice. *Ldlr$^{-/-}$* mice were on chow diet for 30 weeks and developed native atherosclerotic lesions as reported previously[14]. *Mif*-deficient mice (*Ldlr$^{-/-}$ Mif$^{-/-}$*) were used for comparison. Aortic root sections were deparaffinized and rehydrated. For antigen retrieval, slides were boiled in sodium citrate buffer, pH 6.0, 0.05% Tween 20, and blocked with PBS, containing 5% donkey serum and 1% BSA. For staining, slides were incubated at 4 °C with Fluos-msR4M-L1 (5 µM) in blocking buffer. DAPI was used for nuclear counterstain and sections were imaged using a Leica DMi8 fluorescent microscope. The mean fluorescence intensity localized to the aortic vessel wall was quantified via ImageJ, accounting for autofluorescence background signals.

For *Apoe$^{-/-}$* mice (and *Apoe$^{-/-}$Mif$^{-/-}$* as control[66]), cryo-conserved sections of advanced lesions from brachiocephalic artery (BC) were used from mice on Western-type high-fat diet (HFD, 1.25% cholesterol) for 24 weeks. Slides were fixed in ice-cold acetone, rehydrated in PBS, and blocked in PBS/1% BSA, incubated with 500 nM Fluos-msR4M-L1, and analyzed as above.

**Fluos-msR4M-L1 staining and monocyte adhesion in atherosclerotic carotid arteries by multiphoton microscopy**. Monocyte adhesion experiments in atherosclerotic carotid arteries under physiological flow conditions ex vivo have been established[14,75] and were performed by a slight modification of this procedure. Briefly, seven-week-old *Apoe$^{-/-}$* mice were fed a Western-type HFD (0.2% cholesterol) for 12 weeks. The last 3 days before sacrifice, mice were injected with msR4M-L1 (100 µg, once daily) or sterile saline (control). On day 3, arteries were prepared and mounted into an arteriograph chamber. Carotids were flushed with buffer-containing msR4M-L1 (3 µM). Mouse leukocytes isolated from the bone

marrow of msR4M-L1- or vehicle-treated atherogenic *Apoe$^{-/-}$* mice were stained with fluorescent Green CMFDA or Red CMPTX (Thermo Fisher Scientific). After washing with Hank's Balanced Salt Solution (HBSS), stained leukocytes were incubated with 3 µM msR4M-L1 (red) or PBS (green, control) for 1 h at 37 °C. The red- and green-stained cell pools were mixed at a 1:1 ratio and $3 \times 10^6$ cells in 6 mL perfused into the artery of msR4M-L1- or vehicle-treated mice, respectively. Arteries were scanned by MPM using a multispectral TCS SP8 DIVE instrument with filter-free 4TUNE NDD detection module (Leica) and the number of adherent and transmigrated leukocytes determined by scanning multiphoton excitation. Vessel structure (and plaques) were visualized by second harmonic generation (SHG). Image acquisition, visualization, and processing was performed using the LAS X SP8 software package with LAS X 3D visualization and analysis (Leica) and LAS X HyVolution 2 packages from Leica/SVI Huygens, (Laapersveld, NL).

For plaque staining with Fluos-msR4M-L1 in atherosclerotic carotid arteries and aortic roots, Fluos-msR4M-L1 was i.p.-injected into aged atherogenic *Apoe$^{-/-}$* mice (24-week HFD) three days before carotid preparation (50 µg per mouse, once daily). Aortic roots were fresh-frozen and cut into 8 µm-sections and whole-mount carotid arteries prepared for MPM as above. After recording Fluos-msR4M-L1 signals in aortic root sections by MPM, sections were stained by oil Red O (ORO) solution (0.5% in propylene glycol). The Fluos-msR4M-L1-positive area was quantified by ImageJ as a percentage of whole target area (ORO+ for aortic root, plaque area for carotid).

**Proteolytic stability assay**. Mouse plasma was prepared from blood of wildtype C57/BL6 mice by standard procedure. TAMRA-msR4M-L1 dissolved in 1 mM HCl, mixed with PBS, and added to mouse plasma (final concentration 482 nM) and mixtures incubated at 37 °C for various time intervals up to 48 h. Samples were then diluted in 2x Novex Tricine SDS sample buffer (Life Technologies) at a ratio of 1:6, electrophoresed in a 10–20% Tricine gel, and red fluorescent bands directly imaged with an Odyssey® Fc imager. Human plasma was prepared from blood of healthy volunteers by standard procedure. Biotin-6-Ahx-msR4M-L1 was dissolved in PBS and mixed with PBS or human plasma (final concentration 13.7 µM) and solutions incubated for 0.5, 1, 4, or 16 h at 37˚C. Samples were electrophoresed as above, transferred to nitrocellulose, and Biotin-6-Ahx-msR4M-L1 revealed by streptavidin-POD conjugate (Roche Diagnostics, Mannheim, Germany; 1:5000 dilution), using an Odyssey® Fc imager.

**In vivo stability of msR4M-L1 (pharmacokinetics)**. TAMRA-msR4M-L1 was dissolved in sterile physiological saline (0.9% sodium chloride). C57BL/6 J mice were i.p.-injected with a single dose of TAMRA-msR4M-L1 (50 µg per mouse, corresponding to 2.5 mg/kg; one mouse per time point). Mice were sacrificed at 0, 0.5, 1, 2, 4, 6, 8, 24, and 48 h and blood collected by cardiac puncture in EDTA-coated tubes. Plasma was immediately separated by centrifugation at 3000 rpm for 10 min at 4 °C, and placed on ice protected from light. Plasma aliquots were diluted (1:1) with SDS-PAGE sample buffer, boiled, analyzed in 10–20% Tricine SDS-PAGE gels, and red fluorescent bands imaged by Odyssey® Fc imager. Full-size gels are supplied in the Source Data file.

**Cytokine array**. Cytokine/chemokine profiling was performed from plasma samples of msR4M-L1- *versus* vehicle-treated *Apoe$^{-/-}$* mice using mouse cytokine array panel A (R&D Systems, ARY006) according to the manufacturer's instructions. Plasma samples were diluted (1:10) in array buffer; incubated with antibody detection cocktail for 1 h at RT, exposed to the blocked membranes (overnight, 4 °C), membranes washed and incubated with streptavidin-HRP conjugate working solution (30 min, RT). Membranes were developed with Chemi-Reagent Mix and analyzed by Odyssey® Fc imager. The average signal (mean signal intensity) of duplicate spots was quantified by ImageJ.

**In vivo model of atherosclerosis**. *Therapeutic injections of msR4M-L1 and aorta preparation*. Seven-eight-week-old female *Apoe$^{-/-}$* mice were randomly divided into two groups of 11–12 mice each and both groups put on a Western-type HFD (0.21% cholesterol) for 4.5 weeks. Mice develop early-to-intermediate atherosclerotic lesions in this model[44]. One group was i.p.-injected with 50 µg msR4M-L1 dissolved in saline every other day for 4.5 weeks; controls received saline. No toxicity or side effects were noted. At the end of the experiment, mice were sacrificed, blood collected by cardiac puncture and saved for blood cell and lipid measurements and mice transcardially perfused with saline. Hearts, proximal aortas and carotid arteries were prepared and fixed for plaque morphometry and lesion analysis.

*Quantification of plaques and vessel morphometry (ORO and H&E staining)*. Cut heart tissues containing aortic root were embedded in Tissue Tek optimum cutting temperature (OCT) substance (Sakura Finetek, Osaka, Japan) and frozen at −80 °C. Eight-µm sections were prepared for ORO staining and plaque immune cell analysis. The accumulation of macrophages in aortic root lesions was determined by an anti-MAC-2 antibody (1:100 dilution) followed by Cy5-conjugated secondary antibody. Nuclei were visualized with DAPI. The aortic arch was cut, fixed in 4% paraformaldehyde (PFA) and embedded in paraffin. Four-µm sections containing the three branches (brachiocephalic, left common carotid, and left subclavian artery) were prepared and stained with hematoxylin/eosin (H&E)

for vessel morphometry. Images were captured with a Leica DMi8 microscope and quantified using ImageJ.

*Blood cell counts, triglycerides and cholesterol levels.* Blood was collected in EDTA tubes and leukocytes and plasma obtained by centrifugation at $630 \times g$ (10 min, 4 °C). For leukocyte counts, red blood cells (RBC) were depleted by RBC-lysis buffer (BioLegend) at RT, leukocytes washed and suspended in PBS containing 0.5% BSA. Cells were stained with an antibody cocktail comprising APC-Cy-7-conjugated anti-CD45, PE-conjugated anti-CD11b, APC-conjugated anti-CD19, FITC-conjugated anti-CD3, APC-conjugated anti-Ly6C, and PE-conjugated anti-Ly6G (BD Biosciences). Measurements were analyzed using a BD FACSVerse™ flow cytometer and data quantified using FlowJo software. The gating strategy for the blood leukocyte analysis is shown in Supplementary Fig. 24.

Total cholesterol and triglyceride concentrations were measured enzymatically using routine cholesterol fluorometric and triglyceride colorimetric assay kits, respectively (Cayman Chemical Company, Ann Arbor, USA).

**Analysis of human carotid atherosclerotic plaques.** *Patient population, study groups and tissue samples.* Carotid artery tissue samples came from the Munich Vascular Biobank (MVB) and were from patients who underwent carotid endar-terectomy (CEA) in the Department of Vascular and Endovascular Surgery at University Hospital of Technische Universität München. Healthy carotid vessels were obtained from the Forensic Medicine Department. Sample processing for histological analysis was performed by an established procedure[46,76]. Briefly, carotid specimens were fixed in formalin and embedded in paraffin (FFPE), cross-sections of 2–3 µm were prepared and stained with H&E and Elastica-van-Gieson (EVG) to evaluate tissue histomorphology and plaque vulnerability. Atherosclerotic lesions were characterized according to the American Heart Association (AHA) guidelines[77]. All FFPE stable or unstable carotid tissues showed advanced ather-osclerosis (stage V–VII). Fresh tissue samples were processed immediately as detailed below and were rendered stage I-III (early) or stage V-VII (advanced) based on the aforementioned classification after Stary[77]. The study was approved by the local ethical committee of the University Hospital (Committee 2799/10 and 5290/12; "Ethikkommission der Fakultät für Medizin der Technischen Universität München, Munich, Germany) and followed the Guidelines of the World Medical Association Declaration of Helsinki. All patients provided informed consent.

*Quantitative polymerase chain reaction (qPCR).* For qPCR from FFPE sections, total mRNA was isolated from 19 stable and 20 unstable human CEA plaques, and from 4 healthy vessel tissue sections, using High Pure RNA Paraffin kit (Roche) following an established procedure[46]. The mRNA was reverse-transcribed using First Strand cDNA synthesis kit (Thermo Scientific), and qPCR reactions run using ORA™SEE qPCR Green ROX H MIX (HighQu, Germany).

Primers used were: MIF forward, 5′→3′ AGA ACC GCT CCT ACA GCA AGC; MIF reverse, 5′→3′ GGA GTT GTT CCA GCC CAC AT; actin forward, 5′→3′ AGA GCT ACG AGC TGC CTG AC; actin reverse, 5′→3′ CGT GGA TGC CAC AGG ACT.

For qPCR from fresh tissue samples, CEA tissue of nine early versus nine advanced plaque stages (age- and gender-matched), was homogenized in QIAzol lysis reagent (Qiagen, Netherlands) and total RNA isolated using the miRNeasy Mini Kit (Qiagen) according to manufacturer′s instruction. First strand cDNA synthesis was performed using the High-Capacity-RNA-to-cDNA Kit (Applied Biosystems, USA).

Quantitative real-time TaqMan PCR was performed using commercial primers from Thermo Fisher (MIF: Hs00236988_g1; RPLP0: HS00420895_gH). PCRs were run on a QuantStudio5 Cycler (Applied Biosystems, USA). Gene expression was normalized to Rplp0 and quantified with the $2^{\wedge}\Delta\Delta Ct$ method.

*RNA sequencing (RNAseq).* Library preparation was performed using the TruSeq Stranded Total RNA Library Prep Kit with Ribo-Zero (Illumina, Berlin, Germany). Briefly, human carotid artery tissue from was homogenized in QIAzol lysis reagent (Qiagen, Netherlands), total RNA isolated using the miRNeasy Mini Kit (Qiagen, Netherlands) according to manufacturer′s instruction, and RNA integrity number (RIN) determined with the Agilent 2100 BioAnalyzer (RNA 6000 Nano Kit, Agilent). For library preparation 1 µg of RNA was depleted for cytoplasmatic rRNAs, fragmented, and reverse transcribed. A-tailing, adaptor ligation, and library enrichment were performed as described in the high throughput protocol of the TruSeq RNA Sample Prep Guide (Illumina). RNA libraries were assessed for quality and quantity with the Caliper LabChIP GX and the Quant-iT PicoGreen dsDNA Assay Kit (Life Technologies). Libraries were pooled and ran as 150 bp paired-end runs on an Illumina NovaSeq 6000 platform.

*Immunohistochemistry/immunofluorescence staining.* Immunofluorescence staining of human CEA tissues (11 stable, 12 unstable plaques) and nine healthy vessel controls with TAMRA-msR4M-L1 was performed using a similar protocol as for paraffin-embedded specimens from $Ldlr^{-/-}$ mice (see above), except that 1.5 µM of the fluorescent probe was used. For the competition experiment with MIF (54–80), 1.5 µM TAMRA-msR4M-L1 was pre-incubated with 3 µM MIF(54–80) before adding the mixture to the CEA slides. TAMRA-msR4M-L1 staining was corrected for autofluorescence, e.g. at tissue edges. Antibody-based detection of MIF was performed applying the DAB + kit (Abcam, ab64238) following a standard protocol. MIF was detected with the polyclonal goat antibody N-20 (Santa Cruz, sc-16965; 1:100) or protein A-purified IgG from rabbit anti-MIF polyclonal serum Ka565[14]. HRP-conjugated polyclonal rabbit anti-goat immunoglobulin

(DAKO, P0160, 1:1000) or goat anti-rabbit immunoglobulin (Dako REAL™ Detection System, K5001) was used as secondary antibody. Slides were counter-stained with Mayer's hematoxylin and stainings analyzed with a Leica DMi8.

**Statistical analysis.** Statistical analysis was performed using GraphPad Prism version 6, 7, and 8 software. Data are represented as means ± SD. After testing for normality using the D'Agostino-Pearson omnibus normality test, data were ana-lyzed by two-tailed unpaired Student's $t$ test or by two-tailed Mann–Whitney U test, or by one-way ANOVA with Tukey's multiple comparisons test or Kruskal–Wallis test with Dunn's multiple comparisons test, as appropriate. Dif-ferences with $p < 0.05$ were considered to be statistically significant.

**Reporting summary.** Further information on research design is available in the Nature Research Reporting Summary linked to this article.

## Data availability

The data supporting the findings of this study are available within the paper and its Supplementary Information files. The crystal structures of human CXCR4 as used for ectodomain peptide selection in this study are available as publicly available datasets under the accession codes 3ODU, 3OE0, 3OE6, 3OE8, 3OE9, 4RWS. All other data are available from the corresponding authors upon reasonable request. Source data are provided with this paper.

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

## Acknowledgements
This work was supported by Deutsche Forschungsgemeinschaft (DFG) grant SFB1123-A3 to J.B. and A.K., DFG INST 409/209-1 FUGG to J.B., SFB1123-A1 to C.W., SFB1123-Z2 to R.T.A.M., SFB1123-B3 to M.D. and Y.A., SFB1123-B5 to L.M. and by DFG under Germany's Excellence Strategy within the framework of the Munich Cluster for Systems Neurology (EXC 2145 SyNergy—ID 390857198) to J.B., C.W., and M.D. C.W. is Van de Laar Professor of Atherosclerosis and R.B. is Waldemar Von Zedtwitz Professor of Medicine and supported by NIH R01 AR049610. A.H. is supported by a Metiphys scholarship of LMU Munich. We thank Dr. Philipp von Hundelshausen and Dr. Xavier Blanchet for providing recombinant CXCL12, Dr. Robert Kleemann for plaque specimens from $Ldlr^{-/-}$ and $Ldlr^{-/-}$ $Mif^{-/-}$ mice, Mathias Holzner for assistance with the chemotaxis experiments, Carolus Therapeutics, Inc. for providing MIF peptide fragments, Dr. Thomas Hennes for cloning MIF(10xAla), and Simon Besson-Girard for help with the statistical analyses. We thank Dr. Sophie Brameyer and the Biophysics Core Facility at the School of Biology of LMU Munich and Prof. Michaela Smolle at the LMU Biomedical Center for usage of their MST instruments. We thank the mass spectrometry facilities of Technische Universität München (TUM) (Department of Chemistry, Garching; Bavarian Center for Biomolecular Mass Spectrometry, BayBioMS, Freising) for mass spectrometric measurements, Dr. Joana Viola-Söhnlein for help with the Luminex instrument, and Dr. Dorothee Atzler for advice regarding the pharmacokinetic experiments.

## Author contributions
J.B. and A.K. conceived the project, and they designed experiments with contributions from C.Ko., O.E., Y.A., R.T.A.M., L.L., R.B., O.G., and C.W. C.Ko., O.E., C.Kr., D.S., K.H., G.Y., C.Z., M.B., P.B., R.T.A.M., A.H., Y.G., Je.P., W.E.K., S.G., and S.W. conducted the experiments. C.Ko., O.E., C.Kr., G.Y., C.Z., M.B., P.B., A.H., W.E.K., Y.G., and S.W. contributed to data analysis. R.B., L.L., L.M., M.D., H.-H.E., W.E.K., and Ja.P. contributed critical materials. J.B. and A.K. wrote the manuscript with contributions from all authors.

## Funding

## Competing interests
J.B., R.B., and C.W. are co-inventors of patents covering anti-MIF strategies (antibodies, small molecules, and MIF sequence-derived peptides) for inflammatory and cardiovascular diseases. C.Ko., A.K., O.E., and J.B. are co-inventors of a patent application covering MIF-binding CXCR4 ectodomain mimics for inflammatory and cardiovascular diseases. The remaining authors declare no competing interests.
