## [Peer Review File · Nature Communications]

Reviewers' comments:

Reviewer #1 (Remarks to the Author):

The paper from Kontos et al. describes the development and in vitro and in vivo testing of a chimeric molecule that specifically inhibits MIF signalling to CXCR4 without impacting on CXCL12 interaction with CXCR4 or MIF-CD74 signalling. This is a very relevant investigation, concerning the importance of inflammatory and immune function in diseases like atherosclerosis and given the divergent effects the different interactions have on cell function and disease development. The studies are very well performed and build on thorough design and testing of peptides. It also reads very well.

Concerns:

1. In vivo studies are performed by 3-times-a-week injections of 50ug of msR4M-L1 per apoe^{-/-} mouse. In the discussion it is reported that this dose is justified because it establishes relevant inhibitory concentrations but data from PK studies is not shown. These data should be given or such experiments should be performed.
2. In the initial in vitro testing studies, B cells are used because they express CXCR4 regulating B cell migration. What is the effect of msR4M-L1 on B cells in the in vivo studies. Total lymphocytes are not significantly affected although a trend towards reduction is suggested (3070 -> 2347) in n=5 animals. Were B-cell levels in circulation or lymphoid organs changed? Did the treatment impact on B-cell function? Were antibody levels for instance changed?
3. The binding studies using human plaques are somewhat confusing and should be better investigated or at least explained. Higher binding of msR4M-L1 to stable plaques is observed, while binding to unstable or vessels without atherosclerosis was equal. Does this imply that MIF is absent in unstable plaques? And does this not imply that targeting of MIF in plaques is not desirable, as it is only present in stable plaques? Specificity of the binding should be confirmed by competition with unlabelled MIF.
4. The relevance of msR4M-L1 in inhibiting macrophage foam cell formation is briefly touched upon but this would benefit from further work. Studies are done using labelled LDL, but this does not relevantly mimic foam cell formation as this is likely mediated by LDLR-dependent uptake. LDLR dependent uptake is shut down upon lipid loading of cells and thus does not lead to relevant intracellular levels of cholesterol esters. Is LDLR expression affected by the inhibitory peptide? And does it impact on scavenger receptors and uptake of for instance acLDL or oxLDL?
5. Static cell adhesion is studied comparing TNF induced and MIF induced adhesion. While MIF dependent adhesion can be blocked, it is unclear whether MIF dependent adhesion to TNF-activated (since this stimulus seems relevant in in vivo settings) endothelial cells can be blocked by msR4M-L1.

Reviewer #2 (Remarks to the Author):

In this manuscript, the authors describe their work neutralizing a pro-atherogenic chemokine (MIF) by designing a neutralizing receptor that binds MIF but not CXCL12, a different ligand with affinity to the same receptor CXCR4. This selectivity is relevant because CXCL12 is protective, while MIF is atherogenic. The authors employ rational peptide design resulting in remarkable 140-fold selectivity (40nM versus 5uM for MIF and CXCL12, respectively) and then test the drug candidate in vitro, in mice with atherosclerosis and for binding in human plaque. The lesion size reduction is impressive. Overall the work appears to be well done, comprehensive and timely.

The authors state that "CXCR4 mimic msR4M-L1 not only homes to and marks atherosclerotic plaque tissue". I wonder if "homing" is really the right term. A cell expressing the receptor would "home" by actively migrating, but the receptor by itself can only passively enrich or be retained.

For the in vivo study, the drug was injected quite frequently. Can the authors please comment on whether this is a viable application route in patients with chronic atherosclerosis?

Reviewer #3 (Remarks to the Author):

The authors cleverly construct a CXCR4 ectodomain fusion peptide that binds MIF and selectively blocks its functional interaction with the CXCR4 receptor without significantly affecting MIF-CD74 interactions or CXCL12-CXCR4 binding. The concept is novel, and the biological data suggest that a selective MIF inhibitor could have salutary effects in a mouse model of atherogenesis. The principal concern is that the molecular pharmacology of the synthetic peptide inhibitor is insufficiently proven by the biophysical measurements presented in the first two figures.

While the dot blot comparison of L1 peptide binding to CXCL12 vs MIF is a strong qualitative result, the authors go too far in their interpretation of the fluorescence data as evidence of formation of a specific, high-affinity complex with MIF, since fluorescence intensity can be affected by many different effects other than a specific 1:1 binding reaction. Fluorescence polarization, MST and/or NMR would be a more suitable approach. Aggregation behavior that both L1 and L2 exhibit (fig S4) raises serious concerns about the utility of the fluorescence assay for measuring binding affinities. Aggregation could explain differences in the apparent K_d 's for the two CXCR4 ligands, since MIF ($pI \sim 7$) is more likely to be aggregation-prone whereas CXCL12 is a highly soluble protein with only a very weak tendency to form homodimers. MST was used to show that L1 does not interfere with MIF-CD74 binding, but these data are not shown.

The formation of specific interactions between unstructured protein fragments in solution is highly improbable. One of the molecules (L1) may be partly folded, as suggested by the CD data 1e, but there is no evidence of a stable structure from cooperative unfolding or more detailed structural analysis. Structure-function analysis to map the inhibitor binding region of MIF is useful, but it would be more compelling to show that the interaction can be disrupted by mutating one or a few amino acids in the proposed complex.

Specific suggestions:

1. Obtain MST binding data for L1 peptide with MIF and CXCL12, and show these binding curves along with the competition assays described in the first section of the results.
2. Included representative data in fig 2 for at least one of the gray (non-binding) peptides from fig S6 to illustrate the difference between binding and non-binding MIF fragments.
3. Eliminate molecular modeling from the paper, unless 2D NMR spectra are obtained of the L1 peptide in the absence and presence of MIF and CXCL12 showing that a specific complex is formed, which can then be modeled with the benefit of residue-specific chemical shift mapping.

Reviewer #4 (Remarks to the Author):

Following on from the CANTOS Trial (NEJM 2017) - identifying novel anti-inflammatory strategies for treating atherosclerosis are exceedingly topical. In targeting chemokines - a challenge to investigators relates to the promiscuous interactions of these proteins with a multitude of receptors. The authors have built on their important recent observations in relation to macrophage migration inhibitory factor (MIF) and in particular identifying specific and selective binding areas in the MIF structure that are responsible for MIF/CXCR2, MIF/CD74 and MIF/CXCR4 interactions and biological effects.

In this paper they build on this work by designing specific soluble CXCR4 ectodaim mimics to

selectively inhibit MIF/CXCR4 interaction and showed functional inhibition in both in vitro vascular and in vivo atherosclerosis animal models. In addition, and most importantly they showed selectivity. This body of work is important not only in the context of MIF biology but also in the context of the wider chemokine biology field.

In review of the Methods/Results - in general they are appropriate. Minor suggestions and for completeness sake would suggest the authors might include the following :-

1.

In Figure 5d, fluorescently labelled CXCR4 mimic was injected into atherosclerotic mice and its targeting to plaque tissue shown? Can this be quantified? Also, images on the indicated minor background staining to other organs (liver, brain) could be shown as supplemental figure.

2.

The in vitro static arrest data in Figure 5a could be further extended to in vitro flow arrest experiments.

3.

The mass spectrum for the main mimic is shown; the mass spectra for the main other mimic as reported on in the table should also be shown (as supplementary data).

Point-by-point response to the Reviewers' critique

We thank the 4 Reviewers and the Editor for their valuable and very helpful comments on our manuscript. Please find our point-by-point response below. All changes and edits in the revised manuscript are highlighted by red text color for tracking purposes. New figure panels and supplementary figures are indicated by red color letters and red-color figure legends.

Reviewer #1:

General

The paper from Kontos et al. describes the development and in vitro and in vivo testing of a chimeric molecule that specifically inhibits MIF signalling to CXCR4 without impacting on CXCL12 interaction with CXCR4 or MIF-CD74 signalling. This is a very relevant investigation, concerning the importance of inflammatory and immune function in diseases like atherosclerosis and given the divergent effects the different interactions have on cell function and disease development.

The studies are very well performed and build on thorough design and testing of peptides. It also reads very well.

Response: We thank the Reviewer for the positive assessment of our manuscript and the valuable comments.

Concerns

1. In vivo studies are performed by 3-times-a-week injections of 50ug of msR4M-L1 per apoe-/- mouse. In the discussion it is reported that this dose is justified because it establishes relevant inhibitory concentrations but data from PK studies is not shown. These data should be given or such experiments should be performed.

Response: In the previous version of the manuscript, we had determined the proteolytic stability of msR4M-L1 in human plasma in an *ex vivo* setting (incubation of human plasma with biotin-msR4M-L1 as msR4M-L1-derived tracer; estimated 45% of biotin-msR4M-L1 recovered after 16 h of plasma exposure; see Fig. 6a of the initial version of the manuscript) and chosen the dose for the *in vivo* atherosclerosis study based on these experiments and prior state-of-the-art from comparable models, but the Reviewer raises an important point.

As requested by the Reviewer, in revision we have now also performed an *in vivo* stability (PK) experiment. We used a fluorescently labeled analog of msR4M-L1 as tracer following similar approaches by others for PK/biodistribution studies of peptides/proteins (e.g. Kuna et al., Molecular size modulates pharmacokinetics, biodistribution, and renal deposition of the drug delivery biopolymer elastin-like polypeptide, *Sci Rep* 2018). We used the red-fluorescent variant TAMRA-msR4M-L1 in combination with SDS-PAGE and fluorescence imaging for monitoring the stability of this msR4M-L1 analog.

We first tested the proteolytic stability of TAMRA-msR4M-L1 in mouse plasma *ex vivo* (starting concentration: 482 nM). Confirming the previously obtained data with the biotin-msR4M-L1

tracer in human plasma, this experiment showed that approximately 40% of TAMRA-msR4M-L1 (corresponding to approximately 200 nM) could be detected in mouse plasma 48 h after mixing, corresponding to an apparent half-life of ca. 20 h (new Fig. 6a and new Supplementary Fig. 22a).

We next tested whether the tracer signal in a mouse plasma environment would be suitable for quantification purposes and whether it follows a linear dose behavior. New Supplementary Fig. 23 shows that this is the case, that TAMRA-msR4M-L1 follows a linear dose-signal behavior, and that the signal is very sensitive (lower limit of detection of down to 1-3 ng per lane), suggesting that it is suitable in principal.

Scouting experiments in a full *in vivo* setting (injection of 50 µg TAMRA-msR4M-L1 into a C57/BL6 mouse; plasma collection after 24 h) further verified this notion and indicated that measurable amounts of the tracer (>100 ng/mL) can be detected in plasma by SDS-PAGE/fluorescence imaging after an extended *in vivo* period (not shown).

Lastly, a full *in vivo* stability (PK) experiment was performed with 9 mice (time points 0-48 h), plasma samples collected and analyzed by SDS-PAGE. The TAMRA-msR4M-L1 band (monomer and apparent dimer) was visualized by red fluorescence imaging (Odyssey® Fc imager) and quantified using the above established plasma dose calibration curve. New Fig. 6b indicates that measurable concentrations of the CXCR4 mimic appeared in the circulation 1 h after administration (corresponding to approximately 200 ng/mL), as expected for an i.p. leg, and were detectable for up to 48 h (corresponding to approximately 100 ng/mL). We did not determine formal PK parameters using this approach, but the analysis clearly shows that a typical maximum curve as expected for an i.p. leg (plasma peak at 1-2 h) is obtained, that approximately 1-5% of the injected dose was detectable in circulation (corresponding to approximately 200 ng/mL or 50 nM), and that the peptide is relatively stable after reaching the circulation. This is in very good agreement with our *ex vivo* plasma exposure experiments (see above).

These data thus suggested that plasma concentrations between our repetitive injections in the *in vivo* atherosclerosis experiments were maintained at a steady dose range of 20-50 nM, overall in line with the inhibitory concentrations of msR4M-L1 of 30-80 nM as determined in the various *in vitro* and cellular assays (see e.g. Fig. 3d or 4g/i). Moreover, a significant portion of the injected dose enriches in atherogenic lesions (as indicated by the experiments in Fig. 5), suggesting that local tissue concentrations of msR4M-L1 could even be higher than 50 nM.

The *in vivo* stability (PK) experiment thus overall justifies the applied dose. The results and methods sections were re-written accordingly (Results and Methods part of the revised manuscript, page 17-18 and 38, respectively).

2. In the initial in vitro testing studies, B cells are used because they express CXCR4 regulating B cell migration. What is the effect of msR4M-L1 on B cells in the in vivo studies. Total lymphocytes are not significantly affected although a trend towards reduction is suggested (3070 -> 2347) in n=5 animals. Were B-cell levels in circulation or lymphoid organs changed? Did the treatment impact on B-cell function? Were antibody levels for instance changed?

Response: The Reviewer raises a very interesting point. In fact, the analysis of the circulating lymphocyte count is suggestive of a trend towards a reduction of lymphocytes in the treatment group. We acted on the Reviewer's suggestion to specifically study B cells in our model. In several revision experiments, we examined the potential effect of msR4M-L1 treatment on B cells, by measuring total plasma immunoglobins (Igs), anti-oxLDL Igs, spleen B- (as well as T) cell numbers, and splenic IgM- and IgD-positive areas as a surrogate indicator for marginal zone *versus* follicular B cells.

In conjunction, these data do not suggest a major effect of msR4M-L1 treatment on B-cell function, at least not under the conditions of our model of early atherosclerosis. We have not included these data in the revised manuscript, but summarize them below as ‘Reviewer-only’-Figures 1-3.

We first looked at plasma levels of total Igs. Six different Ig sub-classes were measured in the plasma samples obtained after the 4.5-week HFD and treatment period. Plasma Igs did not significantly differ between the vehicle and treatment group in our model of atherosclerosis (**Reviewer-only Figure 1a**). **Reviewer-only Figure 1b** similarly shows that plasma levels of anti-oxLDL IgM or IgG did not differ between both groups.

Reviewer-only Figure 1: msR4M-L1 treatment in an *in vivo* model of early atherosclerosis does not affect total plasma immunoglobulin (Ig) levels or plasma antibody levels against oxLDL. Atherogenic *ApoE*^{-/-} mice on a 4.5-week high-fat diet (HFD) received msR4M-L1 or vehicle injections over the HFD time course. **a** Plasma concentrations of Ig subtypes IgG2b, IgG2c, IgG3, IgM, IgE, and IgA were determined with ProcartaPlex Mouse Antibody Isotyping Panel 2 (“7-Plex”, Invitrogen-Thermo Fisher Scientific) using a Luminex plate reader and following the manufacturer’s recommendations. Plasma samples were diluted 1/40000 and Igs revealed with 50 μ L of antibody magnetic beads. Antigen standards were prepared according to the manual and certificate of analysis. Analysis was done by ProcartaPlex Analyst 1.0 software (ThermoFisher) and the antibody concentration for each isotype calculated using the standards. IgG1 levels were below threshold and could not be reliably derived from the available plasma samples. Plasma samples from 9-10 mice per group were analyzed. Data are reported as means \pm SD; statistical analysis was performed with unpaired T-test. **b** Plasma concentrations of antibodies against oxidized LDL (oxLDL). Anti-oxLDL IgM and anti-oxLDL IgG levels were determined by anti-oxLDL-specific ELISAs using oxLDL-coated Nunc MaxiSorp plates. Plasma was applied at a 1/100 dilution in 2% BSA/PBS. Anti-oxLDL antibodies were revealed with horseradish-peroxidase (HRP)-labeled antibodies against IgG (#ab6789, Abcam) and IgM (#62-6820, ThermoFisher Scientific) diluted 1/500 in 2% BSA/PBS. Signals were developed by TMB-substrate solution and chemiluminescence reaction using an EnSpire microplate reader (PerkinElmer). The signal of uncoated wells served as a background control for each individual plasma sample and was subtracted from the chemiluminescence signal of the respective coated wells.

We next analyzed B-cell numbers in the spleens prepared at the end of the 4.5-week period. Splenic B cells were quantified both by CD19 qPCR and immunohistochemistry staining for B220. **Reviewer-only Figure 2** shows that neither CD19 mRNA levels nor B220-positive splenic areas significantly differed between groups. Also, T cells as measured by Cd3e-positivity were not significantly different between both groups.

Reviewer-only Figure 2: msR4M-L1 treatment in an *in vivo* model of early atherosclerosis does not affect splenic B and T cell numbers. Atherogenic *Apoe*^{-/-} mice on a 4.5-week high-fat diet (HFD) received msR4M-L1 or vehicle injections over the HFD time course. **a** Determination of total splenic B cells by measuring CD19 mRNA levels using qPCR. mRNA was extracted from frozen spleen tissue using a routine protocol and reverse-transcribed into cDNA using the First Strand cDNA Synthesis Kit (Thermo Fisher Scientific). qPCR was performed using ORA™ qPCR Green ROX L Mastermix (highQu GmbH, Germany) in a Rotorgene Q (Qiagen, Netherlands) and CD19-specific mouse primer pairs (Metabion Int.; *Cd19* forward primer: GGT ACC GCC ACC ATG GCA CCT CCT CGC CTC CTC TTC, *Cd19* reverse primer: AAG CTT GCC ACC TGA GGA TCA CCT GGT GC). Relative mRNA levels were calculated using the $\Delta\Delta C_t$ method with actin (forward primer: GGA GGG GGT TGA GGT GTT; reverse primer: GTG TGC ACT TTT ATT GGT CTC AA) as a housekeeping gene. Results are depicted as mRNA fold change and were normalized to mRNA levels of non-injected *Apoe*^{-/-} control mice. **b** Determination of splenic B and T cells by B220 and CD3e immunofluorescence, respectively. Immunohistochemistry (IHC) was performed on 8 μ m spleen cryosections. Following blocking in 5% donkey serum, 1% BSA in PBS, sections were incubated with primary antibody as recommended by the provider. For biotin-streptavidin-based IHC, endogenous biotin was additionally blocked using an avidin/biotin blocking kit (Vector Laboratories). The following antibody combinations were applied: primary hamster anti-mouse Cd3e antibody (dilution 1/100, #553058, BD Biosciences, USA) in combination with a primary biotinylated rat anti-mouse B220/CD45R (dilution 1/200, #553086, BD Biosciences, USA), followed by secondary anti-hamster Cy3 (dilution 1/300, #127-165-160, Jackson ImmunoResearch Laboratories) in combination with FITC-Avidin (dilution 1/500, #434411, Thermo Fisher Scientific, USA). DAPI was used for nuclear counterstain. The sections were mounted with Fluoromount™ (Sigma-Aldrich) and images recorded with a DMI8 fluorescent microscope (Leica) and quantitated using ImageJ software. Scale bar: 500 μ m.

Furthermore, we determined splenic B-cell sub-types by quantifying the IgD-, IgM-, and IgG-positive areas. IgD is a marker for follicular B (FOB) cells in the spleen, which at the same time are low in IgM; marginal zone (MZB) cells are IgM^{high} and IgD^{low}. There was an increase in IgM-positive area in the msR4M-L1-treated compared to the vehicle group, but IgD staining did not differ between both groups (**Reviewer-only Figure 3**). IgG staining also did not differ between both groups.

Reviewer-only Figure 3: msR4M-L1 treatment in an *in vivo* model of early atherosclerosis does not affect antibody profiles/subtypes of splenic B cells. Atherogenic *Apoe*^{-/-} mice on a 4.5-week high-fat diet (HFD) received msR4M-L1 or vehicle injections over the HFD time course. **a** Determination of splenic B-cell subtypes by IgM, IgD, and IgG immunofluorescence staining. Immunohistochemistry (IHC) was performed on 8 μm spleen cryosections. Following blocking in 5% donkey serum, 1% BSA in PBS, sections were incubated with primary antibody as recommended by the provider. For biotin-streptavidin-based IHC, endogenous biotin was additionally blocked using an avidin/biotin blocking kit (Vector Laboratories). The following antibody combinations were applied: Texas-Red-labelled goat anti-mouse IgM (dilution 1/100, #1021-07, Southern Biotech, USA) with biotinylated primary rat anti-mouse IgD antibody (dilution 1/100, #1120-08, Southern Biotech, USA), followed by incubation with FITC-avidin (dilution 1/500, #434411, Thermo Fisher Scientific, USA); incubation with biotinylated anti-mouse IgGκ binding protein (dilution 1/50, #sc-516142, Santa Cruz, USA), followed by FITC-avidin (dilution 1/500, #434411, Thermo Fisher Scientific, USA). After washing and DAPI nuclear counterstain, sections were imaged using a Leica DMI8 microscope. Scale bar: 500 μm. **b** Quantification according to **a**. Spleen sections from 12 and 10 mice per group were quantified as indicated. Each data point represents an individual mouse. The IgG⁺/IgD⁺/IgM⁺ splenic area was quantified using image J and values normalized to the mean of saline-injected control animals. IgD/IgM ratios did not differ between groups (not shown).

Splenic B cells are predominantly of the B2 subtype, which generally are considered to be proatherogenic. However, the IgM^{high} IgD^{low} MZB cells have more recently also been assigned an atheroprotective role by controlling T follicular helper cells (Nus et al., *Nat Med* 2017). The observed small increase in IgM^{high} IgD^{low} MZB cells in the treatment group could therefore be an indication that the atheroprotective effect of msR4M-L1 treatment may partly be due to an effect on B cells. In this regard, our observation that circulating levels of the B-cell (and T-cell) chemokine CXCL13 were found to be reduced in the msR4M-L1-treated group (see Supplementary Fig. 24) is interesting, but further studies would be needed to study potential causative links.

As all other B cell parameters measured, including circulating IgM and anti-oxLDL IgM levels, did not differ between the treatment and vehicle group, we would overall conclude, that the data do not support an effect of msR4M-L1 on B-cell function in the 4.5-week model of early atherosclerosis applied in our study. However, this would not rule out effects of this inhibitor in more advanced models of atherosclerosis. In fact, our previous study that revealed a link between MIF and B cells in atherosclerotic lesion formation in the brachiocephalic artery and abdominal aorta was conducted in *ApoE*^{-/-} models of 16-24 weeks of Western diet (Schmitz et al., *FASEB J* 2018). Nus et al. (*Nat Med* 2017) observed the atheroprotective effect of MZB cells in *Ldlr*^{-/-} models of 8-16 weeks of Western diet. It could be hypothesized that potential effects of msR4M-L1 (and thus the MIF/CXCR4 axis) on B cells may only become apparent in more advanced stages of atherosclerosis, a notion that would deserve more detailed future studies and that would be beyond the scope of this manuscript.

We have thus incorporated our current findings as Reviewer-only figures in this response letter, as we feel they are somewhat preliminary in the above-discussed context. Should the Reviewer (and Editor) recommend to move the B-cell data into the Supplement of the manuscript, we are happy to do so.

3. The binding studies using human plaques are somewhat confusing and should be better investigated or at least explained. Higher binding of msR4M-L1 to stable plaques is observed, while binding to unstable or vessels without atherosclerosis was equal. Does this imply that MIF is absent in unstable plaques? And does this not imply that targeting of MIF in plaques is not desirable, as it is only present in stable plaques? Specificity of the binding should be confirmed by competition with unlabeled MIF.

Response: We thank the Reviewer for raising this important point. MIF is indeed *not* absent in unstable plaques and targeting of MIF in plaques is desirable. According to the data presented in the initial version of the manuscript, MIF content in unstable plaques appeared lower in unstable compared to stable plaques. The Reviewer is right that this was somewhat counter-intuitive and that these data should be better investigated. In fact, unstable plaque specimens are a difficult tissue type to examine, e.g. due to the large areas of necrotic core or a sometimes-poor tissue integrity of some of the surgical specimens. We assume that different cellularity degrees in the stable *versus* unstable plaques tissues may have affected some of the results and led to an under-estimation of MIF content in unstable plaque material.

In revision and following the Reviewer's critique, we have investigated the human plaque data in more detail. We went back and reanalyzed all CEA specimens and reevaluated the data. New and additional specimens were analyzed and MIF quantification now is primarily based on its expression values on mRNA level. We initially performed qPCR from mRNAs extracted from paraffin-embedded sections, comparing 19 stable plaques with 20 unstable plaques and 4 healthy vessel tissues (**new Fig. 6k**). This analysis showed that MIF mRNA levels were markedly upregulated in stable AND unstable plaques compared to healthy vessel tissue, while not significantly differing between stable and unstable plaque phenotypes. We further confirmed this result by RNAseq of 6 stable and 5 unstable fresh tissue plaque specimens (**new Fig. 6m**). To further confirm the MIF/plaque phenotyping, we also performed qPCR from fresh tissue specimens. This procedure was more recently introduced at the Munich Vascular

Biobank and yields higher mRNA yields of better quality (Pauli and Mägdefessel, unpublished) and also allows us to further differentiate atherosclerotic stages. MIF mRNA analysis of 9 early *versus* 9 advanced fresh tissue plaque specimens (for representative example of these tissues see new Supplementary Fig. 25) further confirmed the above results and showed that MIF mRNA levels did not significantly differ between atherosclerotic plaque stages (new Fig. 6l). Together, the mRNA analyses showed that MIF is markedly upregulated in atherogenesis compared to healthy vessel tissue, but also suggests that total MIF levels do not significantly differ between the disease stages. This does not exclude the possibility that MIF expression levels may differ between distinct atherogenic cell types such as ECs, macrophages, foam cells, and VSMCs, and their various subtypes in an atherosclerotic plaque in a context-specific manner (e.g. Burger-Kentischer et al., *Circulation* 2002; Chen et al., *Circulation* 2015).

Based on the MIF mRNA data, we then studied selected specimens with a stable and unstable plaque phenotype, as well as healthy vessel-specimens, and examined MIF protein levels in a side-by-side comparison between conventional anti-MIF Ab/DAB staining-based immunohistochemistry and MIF detection by a TAMRA-labeled CXCR4 mimic probe. These data are included in the revised manuscript as new Figs. 6n-p, confirm the mRNA expression data, and show that TAMRA-msR4M-L1 positivity parallels the IHC signal obtained by anti-MIF Ab staining. Both probes show marked and comparable staining in stable and unstable sections, while only marginal staining is visible in healthy vessel tissue. MIF protein detection by TAMRA-msR4M-L1 also was used to quantify the signal in a larger cohort (11 stable plaques, 12 unstable plaques, and 9 healthy vessels). New Supplementary Fig. 27a-b shows that this data is in line with the mRNA quantifications and side-by-side comparison and reveals marked TAMRA-msR4M-L1 positivity in stable and unstable plaques, which did not significantly differ from each other, but was markedly higher than the signal in healthy vessel tissues.

The red-fluorescent TAMRA label was chosen for the revision experiments instead of the Fluos label, as TAMRA gave lower autofluorescence background signals, and thus gave robust readings, even in less homogeneous surgical tissue specimens with high lipid and acellular portions.

However, we also followed the valuable advice of the Reviewer to further test the specificity of the binding by a competition experiment. We chose MIF peptide fragment 54-80, which in the biochemical binding experiments, was identified to be part of the core msR4M-L1 binding site within MIF (see Figure 2d). Moreover, a new dot blot experiment performed in revision, also in response to a point raised by Referee #3, indicated that MIF(54-80) efficiently competes with binding between full-length MIF and msR4M-L1 (new Fig. 2h). Here, we used this MIF peptide fragment to compete TAMRA-msR4M-L1-based binding to MIF protein expressed in stable plaque sections. New Supplementary Fig. 27c-d shows, and thus further confirms, the specificity of the fluorescently labelled CXCR4 mimic probe, with MIF(54-80) being able to reduce the TAMRA-msR4M-L1 signal by 50-60%.

The results and methods sections were re-written accordingly (Results and Methods part of the revised manuscript, page 18-19 and 41, respectively).

4. The relevance of msR4M-L1 in inhibiting macrophage foam cell formation is briefly touched upon but this would benefit from further work. Studies are done using labelled LDL, but this does not relevantly mimic foam cell formation as this is likely mediated by LDLR-dependent uptake. LDLR dependent uptake is shut down upon lipid loading of cells and thus does not lead to relevant intracellular levels of cholesterol esters. Is LDLR expression affected by the inhibitory peptide? And does it impact on scavenger receptors and uptake of for instance acLDL or oxLDL?

Response: The Reviewer makes an important point, which we have addressed by additional experiments testing the effect of msR4M-L1 on oxLDL uptake and by testing its effect on LDLR expression/internalization.

New Fig. 4a and 4b show the effect of the CXCR4 mimic on MIF-elicited uptake of Dil-oxLDL by human monocyte-derived macrophages. The effect was overall comparable to the effect on native Dil-LDL, was even more pronounced, as a complete blockade of MIF-mediated activation of oxLDL uptake is already seen at a 3-fold molar excess of msR4M-L1 over MIF, compared to a 15-fold excess needed to reach a significant reduction of MIF's effect on LDL uptake. We also verified that activation of oxLDL uptake by MIF is CXCR4-dependent, at least in part, as it is partially blocked by AMD3100 (**new Supplementary Fig. 18**).

We have not further studied the specific scavenger receptor involved, but speculate that there could be an effect of the MIF/CXCR4 axis on scavenger receptor clustering and/or internalization, as e.g. Wong et al. recently showed that certain chemokines including the CXCL12/CXCR4 axis promote CD36 clustering via an integrin/actin remodeling mechanism and macrophage foam cell formation (Wong et al., Chemokine signaling enhances CD36 responsiveness toward oxidized low-density lipoproteins and accelerates foam cell formation. *Cell Rep*, 2016). We feel that elucidating these pathways in detail would be beyond the scope of the current manuscript. In fact, while a general link between MIF and oxLDL-related atherogenic effects has been suggested already many years ago (Atsumi et al., *Cytokine* 2000; Burger et al., *Circulation* 2002; Schober et al., *Circulation* 2004), the specific mechanisms and signaling pathways have remained unexplored.

We also thank the Reviewer for bringing up the issue of foam cell formation by (native) LDL. We agree with the Reviewer that the oxLDL/scavenger receptor axis is the main pathway contributing to foam cell formation in atherosclerotic lesions and our new revision data in Fig. 4a and b show that msR4M-L1 blocks this pathway. However, some studies also imply scavenger receptor-independent pathways. For example, studies in *Cd36/Sra* double knockout mice have observed foam cell formation in *Cd36*- and *Sra*-deficient cells/lesions (Manning-Tobin et al., Loss of SR-A and CD36 activity reduces atherosclerotic lesion complexity without abrogating foam cell formation in hyperlipidemic mice. *Arterioscler Thromb Vasc Biol*, 2009; Moore et al., Loss of receptor-mediated lipid uptake via scavenger receptor A or CD36 pathways does not ameliorate atherosclerosis in hyperlipidemic mice. *J Clin Invest*, 2005). Moreover, Dan Rader's group recently suggested that sortilin-mediated uptake of *native* LDL into macrophages may be an additional mechanism of foam cell formation and contributor to atherosclerosis development (Patel et al., Macrophage sortilin promotes LDL uptake, foam cell formation, and atherosclerosis. *Circ Res*, 2015).

In conjunction, these studies imply some contribution of the LDL/LDLR pathway to foam cell formation. In the initial version of the manuscript, we had focused on a Dil-LDL uptake assay (i.e. "native LDL-driven foam cell formation assay"), which was previously developed by the Gleissner/Runz group in Heidelberg as a high-throughput platform to profile clinical specimens from CAD patients (Domschke et al., Systematic RNA-interference in primary human monocyte-derived macrophages: a high-throughput platform to study foam cell formation. *Sci Rep*, 2018). One key finding of the study was that Dil-LDL uptake by macrophages was dependent on the MIF/CXCR4- but *not* the CXCL12/CXCR4-axis. This foam cell assay therefore appeared valuable to us to specifically monitor a MIF/CXCR4-driven atherogenic event. We would therefore like to keep the Dil-LDL uptake data in the manuscript (now Fig. 4c and new Supplementary Fig. 19 of the revised manuscript) in addition to the newly added Dil-oxLDL uptake data, and hope that the Reviewer would agree with this.

Together, the above revision experiments suggest that MIF promotes both oxLDL and LDL uptake by macrophages and that msR4M-L1 potently blocks both of these processes.

Moreover, we followed up on the Reviewer's advice and asked "whether LDLR expression was affected by the inhibitory peptide". We investigated the effect of msR4M-L1 on LDLR surface expression in monocyte-derived macrophages and its potential effect on MIF-mediated LDL/LDLR internalization applying a FACS-based assay set-up. The data indicate that MIF reduces LDLR cell surface levels, i.e. promotes LDL/LDLR internalization, and that this effect is at least partially reversed by msR4M-L1. In line with this observation, a comparable effect is seen on CXCR4 cell surface levels (**Reviewer-only Figure 4**). This might suggest that

MIF/CXCR4-signaling correlates with the LDL/LDLR endocytosis process, e.g. by signaling crosstalk, or might even imply that CXCR4 and LDLR colocalize/complex and are subjected to a joint internalization event.

We feel that these studies warrant further in-depth (and mechanistic) exploration, which are beyond the scope of the current manuscript, and would thus suggest to not include these data in the manuscript. We have therefore prepared them as **Reviewer-only Figure 4** below. Should the Reviewer and the Editor think that we should include these data in the Supplementary part of the manuscript, we are happy to do so.

Reviewer-only Figure 4: Effect of msR4M-L1 on LDL receptor (LDLR) (a) and CXCR4 (b) cell surface expression and endocytosis in human monocyte-derived macrophages. LDLR and CXCR4 cell surface levels were measured by flow cytometry. Experiments were performed in the presence of LDL and the effect of msR4M-L1 tested in the context of MIF stimulation. Cells were preincubated with 80 nM MIF in the presence or absence of 400 nM msR4M-L1 in culture medium (RPMI 1640-GlutaMax medium containing 100µg/mL penicillin, 100 µg/mL streptomycin, and 0.2% BSA) for 15 min at 37°C and subsequently exposed to LDL (25 µg/mL) for 15 min. After washing with PBS, cells were collected and incubated with anti-hLDLR-APC (1:100, R&D Systems, FAB2148A) and anti-CXCR4-PE (1:100, R&D Systems, FAB170P) at 4°C for 1 h. MIF stimulation enhances LDL-triggered internalization of LDLR and that of CXCR4 (see shifts in Q3 from 15.1% to 8.99% and from 39% to 34.5%, respectively) and both effects are reversed by blockade of the MIF/CXCR4 axis with msR4M-L1 (see shifts in Q3 from 8.99% to 12.6% and from 34.5% to 48.7%, respectively). The data shown are from one of two experiments performed.

5. Static cell adhesion is studied comparing TNF induced and MIF induced adhesion. While MIF dependent adhesion can be blocked, it is unclear whether MIF dependent adhesion to TNF-activated (since this stimulus seems relevant in in vivo settings) endothelial cells can be blocked by msR4M-L1.

Response: Thank you for this important point. In revision, we have performed the suggested experiment and now include static adhesion data using TNF-primed endothelial cells in the revised manuscript. msR4M-L1 fully blocks static monocyte adhesion on TNF-activated endothelial monolayers similar to its effect in a set-up with non-preactivated ECs (**new Fig. 5b**).

Reviewer #2:

General

In this manuscript, the authors describe their work neutralizing a pro-atherogenic chemokine (MIF) by designing a neutralizing receptor that binds MIF but not CXCL12, a different ligand with affinity to the same receptor CXCR4. This selectivity is relevant because CXCL12 is protective, while MIF is atherogenic. The authors employ rational peptide design resulting in remarkable 140-fold selectivity (40nM versus 5uM for MIF and CXCL12, respectively) and then test the drug candidate in vitro, in mice with atherosclerosis and for binding in human plaque. The lesion size reduction is impressive. Overall the work appears to be well done, comprehensive and timely.

Response: Thank you for the positive assessment of our manuscript and your helpful comments.

Specific points

1. *The authors state that “CXCR4 mimic msR4M-L1 not only homes to and marks atherosclerotic plaque tissue”. I wonder if “homing” is really the right term. A cell expressing the receptor would “home” by actively migrating, but the receptor by itself can only passively enrich or be retained.*

Response: We thank the Reviewer for bringing up this point and apologize for the confusing term. The Reviewer is right. The use of the word “homing” in the context of a soluble receptor mimic is not correct and in fact confusing. In the revised manuscript, we have eliminated the terms “home to” and “mark” and have now used the term “enriched” as suggested by the Reviewer (e.g. revised text on page 7 and page 17).

2. *For the in vivo study, the drug was injected quite frequently. Can the authors please comment on whether this is a viable application route in patients with chronic atherosclerosis?*

Response: Thank you for raising this important point regarding future translational applications of the peptide or its improved mimetics.

For our *in vivo* atherosclerosis study, we chose an i.p. application route, as this is a common application route for compounds used in atherosclerosis mouse models. Moreover, the i.p. injections represent a relatively mild burden for the animals and ensure reproducible compound administration. The 3x-per-week application regimen (50 µg/mouse corresponding to 2.5 mg/kg each) was based on: i) our prior experience with experimental mice, in which we had studied the effect of e.g. anti-MIF antibodies on MIF's role in inflammation and atherosclerosis (Schober et al., *Circulation* 2004; Burger-Kentischer et al., *Atherosclerosis* 2006) and ii) an estimation of msR4M-L1 stability in plasma and *in vivo*, aiming at maintaining relatively stable circulating peptide levels over the course of the 4.5-week experiment. Please also see our revision experiments regarding the *in vivo* stability/pharmacokinetics of msR4m-L1, which we have now included in the revised manuscript in response to a comment from Reviewer #1 (**new Fig. 6a and 6b and new Supplementary Fig. 23**). Together, this suggested that the chosen 3x-per-week application regimen led to msR4M-L1 plasma levels at concentrations in the range of (or above) MIF levels as they occur in plasma in models of inflammation or atherosclerosis (i.e. 5-200 ng/ml or 0.4-16 nM). The msR4M-L1 plasma levels (i.e. 20-50 nM) also are in line with the inhibitory concentrations (30-80 nM) as determined in the *in vitro* and cell assays (see e.g. Figs. 3 and 4).

We agree this is no viable application route for patients in general and those with atherosclerotic disease in particular. Any such application would have to be i.v., s.c., or ideally oral, and would need to be based on further optimized and stabilized analogs of msR4M-L1. Also, any such application would be guided by clinical studies such as the recent CANTOS trial, in which several thousand cardiovascular patients were administered the anti-IL-1β antibody canakinumab. In CANTOS, 50-300 mg of the IL-1 antibody canakinumab was subcutaneously administered, initially every 2 weeks, and then every 3 months for a total of 48 months (Ridker et al., *NEJM* 2017).

The CXCR4 mimic used in our current study should be regarded as a proof-of-concept peptide that needs to undergo further rounds of peptide optimization and stabilization, as mentioned above and also discussed in the manuscript. A rough comparison between our peptide and clinically applied proteins such as canakinumab, indicates that the msR4M-L1 doses administered in our atherosclerosis model are higher than the canakinumab doses used in CANTOS, but are in a dose range of similar order of magnitude. Thus, provided the medicinal chemistry to further improve the activity and stability of msR4M-L1 will be successful, the data from our proof-of concept model are generally in line with a development route.

Following the Reviewer's comment, we have extended our discussion to account for this important point. Because of the numerous well-known risks inherent to any drug development pipeline, we have tried to phrase this discussion very carefully though and now state on page 25 in the Discussion of the revised manuscript: "Of note, despite the many inherent differences between the experimental atherosclerosis mouse model applied herein and future clinical applications and the numerous unforeseeable risks along any drug development pipeline, the *in vivo* msR4M-L1 dose and application frequency used in our study overall seems compatible with an optimization path towards an efficacious drug.....".

Reviewer #3:

General

The authors cleverly construct a CXCR4 ectodomain fusion peptide that binds MIF and selectively blocks its functional interaction with the CXCR4 receptor without significantly affecting MIF-CD74 interactions or CXCL12-CXCR4 binding. The concept is novel, and the biological data suggest that a selective MIF inhibitor could have salutary effects in a mouse model of atherogenesis. The principal concern is that the molecular pharmacology of the synthetic peptide inhibitor is insufficiently proven by the biophysical measurements presented in the first two figures.

Response: We thank the Reviewer for the overall positive feedback on our study. We address your valuable comments about the molecular pharmacology as detailed point-by-point below.

1. While the dot blot comparison of L1 peptide binding to CXCL12 vs MIF is a strong qualitative result, the authors go too far in their interpretation of the fluorescence data as evidence of formation of a specific, high-affinity complex with MIF, since fluorescence intensity can be affected by many different effects other than a specific 1:1 binding reaction. Fluorescence polarization, MST and/or NMR would be a more suitable approach. Aggregation behavior that both L1 and L2 exhibit (fig S4) raises serious concerns about the utility of the fluorescence assay for measuring binding affinities. Aggregation could explain differences in the apparent K_d 's for the two CXCR4 ligands, since MIF ($pI \sim 7$) is more likely to be aggregation-prone whereas CXCL12 is a highly soluble protein with only a very weak tendency to form homodimers. MST was used to show that L1 does not interfere with MIF-CD74 binding, but these data are not shown.

Response: We thank the Reviewer for raising this important point. In revision, we have followed the Reviewer's recommendation and performed both fluorescence polarization (FP) and microscale thermophoresis (MST) experiments. These are now shown in the revised manuscript in new Figs. 1f-i and were added to revised Table 1. These data confirm the fluorescence intensity measurements and provide app. K_D values between msR4M-L1 and MIF of 24 nM or 11 nM as measured by FP, and 77 nM as determined by MST, while no binding to CXCL12 was observed (app. $K_D > 2.5 \mu M$). Thus together, fluorescence spectroscopy, FP, and MST suggest a binding affinity between msR4M-L1 and MIF in the 11 – 80 nM range, while no binding was seen for CXCL12 (app. $K_{DS} > 2.5 \mu M$). In our manuscript text revisions, we also have down-toned the conclusions drawn from the fluorescence intensity measurements (corresponding text sections on page 8-10 of the revised manuscript).

We would like to kindly mention that, while msR4M-L1 and -L2 showed some self-association behavior, as shown in Suppl. Fig. 4 (now Suppl. Fig. 6 of the revised manuscript) and as discussed by the Reviewer above, the concentrations used in the fluorescence spectroscopy experiments were >10-fold lower than the app. K_D measured for the self-association of msR4M-L1 (10 nM vs. 142 nM). Experiments were thus performed in a range 'far away' from potential self-association confounders.

The MST data on MIF/CD74 binding as well as FP experiments on MIF/CD74, which we also performed in the revision, are now shown in the revised manuscript in new Supplementary Fig. 9. Both methodologies confirm that msR4M-L1 does not affect the binding between MIF and CD74 (FP: app. K_D = 114.4±47.0 nM vs. 89.4±55.3 nM; P = ns; MST: app. K_D = 33.9±5.0 nM vs. 34.5±13.1 nM; P = ns) and are overall in line with the previously published MIF/CD74 binding affinity (Leng et al., *J Exp Med* 2003).

2. The formation of specific interactions between to unstructured protein fragments in solution is highly improbable. One of the molecules (L1) may be partly folded, as suggested by the CD data 1e, but there is no evidence of a stable structure from cooperative unfolding or more detailed structural analysis. Structure-function analysis to map the inhibitor binding region of MIF is useful, but it would be more compelling to show that the interaction can be disrupted by mutating one or a few amino acids in the proposed complex.

Response: The Reviewer brings up another important point. In revision, we have followed the Reviewer's advice and have generated and studied several core binding region mutants, mutating residues in the binding region of both msR4M-L1 and MIF.

Based on an alanine-scanning analysis (new Supplementary Fig. 12), we suspected that the aromatic Phe, Trp, and Tyr residues in position 102, 103, 104, 107, 189, 190, and 195 could be determinants of msR4M-L1 binding to MIF. Accordingly, mutants/analogs [A¹⁰², A¹⁰³, A¹⁰⁴, A¹⁰⁷, A¹⁸⁹, A¹⁹⁰, A¹⁹⁵]-msR4M-L1 or msR4M-L1(7xAla) and analog [A¹⁰⁴, A¹⁰⁷]-msR4M-L1 or msR4M-L1(2xAla) were synthesized (overview in new Fig. 2b) and MIF binding behavior tested by fluorescence spectroscopy. New Fig. 2f and new Supplementary Fig. 13a show that these mutations led to a more or less complete disruption of binding between MIF and msR4M-L1. We also performed CD spectroscopy on these analogs. In line with a role for the aromatic residues, both analogs exhibited reduced conformational order and increased random coil contents in the following order regarding random coil content: msR4M-L1(7xAla) > msR4M-L1(2xAla) > msR4M-L1 (new Supplementary Fig. 13b).

We also generated a mutant of MIF in which several residues of the core binding region were mutated to Ala (MIF(10xAla)). This mutation also led to an abrogation of msR4M-L1/MIF binding (new Fig. 2g).

Together, the experiments introducing mutations of amino acids in the supposed core binding region of either MIF or msR4M-L1 support a specific interaction between MIF and msR4M-L1.

Furthermore, we performed a binding competition experiment, in which we tested the effect of MIF peptide fragment 54-80. The binding experiment in Fig. 2c of the initial manuscript (now Fig. 2d in the revised manuscript) had provided support of the notion that sequence 54-80 is part of the MIF binding region of msR4M-L1. Binding competition as analyzed by dot blot now also indicates that MIF(54-80) markedly reduces the binding between TAMRA-msR4M-L1 and full-length MIF, respectively (new Fig. 2h).

Please note that, based on this latter experiment and also in response to a comment by Referee #1, we also used the MIF(54-80) competition approach to verify the specificity of TAMRA-msR4M-L1 binding to MIF localized in human CEA plaque specimens (new Supplementary Fig. 27c-d).

Specific suggestions

3. Obtain MST binding data for L1 peptide with MIF and CXCL12, and show these binding curves along with the competition assays described in the first section of the results.

Response: Following the Reviewer's suggestion and as also already briefly described under point 1 above, we have performed MST experiments. The data confirm that msR4M-L1 binds to MIF with low nanomolar affinity (determined app. $K_D = 77$ nM), but does not appreciably bind to CXCL12 (determined app. $K_D > 3$ μ M). Additionally, we also performed FP to obtain binding data for msR4M-L1 with MIF and CXCL12. These measurements further confirmed the specific interaction between msR4M-L1 and MIF, but not CXCL12 (msR4M-L1/MIF: determined app. $K_D = 24$ nM or 11 nM using Fluos-msR4M-L1 or Alexa-MIF as tracer, respectively; msR4M-L1/CXCL12: determined app. $K_D > 2.5$ μ M). Thus together, FP and MST confirm a good binding affinity between msR4M-L1 and MIF (10 – 80 nM range), while no binding was seen for CXCL12 (app. $K_{DS} > 2.5$ μ M). The new data are now shown in new Fig. 1f-i of the revised manuscript.

4. Include representative data in fig 2 for at least one of the gray (non-binding) peptides from fig S6 to illustrate the difference between binding and non-binding MIF fragments.

Response: Thank you for this suggestion. We are now showing the data for the non-binding (gray) peptide MIF(6-23) in new Fig. 2e. Additionally, we are showing the data for another non-binding MIF peptide (MIF(62-80)) in new Supplementary Fig. 11.

5. Eliminate molecular modeling from the paper, unless 2D NMR spectra are obtained of the L1 peptide in the absence and presence of MIF and CXCL12 showing that a specific complex is formed, which can then be modeled with the benefit of residue-specific chemical shift mapping.

Response: We agree that the molecular modeling data on their own would be too preliminary. Following the Reviewer's recommendation, we have eliminated these data.

Reviewer #4:

General

Following on from the CANTOS Trial (NEJM 2017) - identifying novel anti-inflammatory strategies for treating atherosclerosis are exceedingly topical. In targeting chemokines - a challenge to investigators relates to the promiscuous interactions of these proteins with a multitude of receptors. The authors have built on their important recent observations in relation to macrophage migration inhibitory factor (MIF) and in particular identifying specific and selective binding areas in the MIF structure that are responsible for MIF/CXCR2, MIF/CD74 and MIF/CXCR4 interactions and biological effects.

In this paper they build on this work by designing specific soluble CXCR4 ectodomain mimics to selectively inhibit MIF/CXCR4 interaction and showed functional inhibition in both in vitro vascular and in vivo atherosclerosis animal models. In addition, and most importantly they showed selectivity. This body of work is important not only in the context of MIF biology but also in the context of the wider chemokine biology field.

Response: We thank the Reviewer for the positive feedback on our study.

In review of the Methods/Results - in general they are appropriate. Minor suggestions and for completeness sake would suggest the authors might include the following:-

1. In Figure 5d, fluorescently labelled CXCR4 mimic was injected into atherosclerotic mice and its targeting to plaque tissue shown? Can this be quantified? Also, images on the indicated minor background staining to other organs (liver, brain) could be shown as supplemental figure.

Response: Thank you for bringing up this point. In the revised manuscript, we now include a quantification of the CXCR4 mimic signal in the carotid plaques (new Fig. 5g). Additionally, we analyzed the aortic root and now also show images of root plaque-localized Fluos-msR4M-L1 as well as corresponding quantification in new Fig. 5g and 5h.

[REDACTED]

2. The *in vitro* static arrest data in Figure 5a could be further extended to *in vitro* flow arrest experiments.

Response: Thank you for this suggestion. In revision, we have performed monocyte adhesion studies under flow conditions. These data are now included in the revised manuscript as new Fig. 5c. The data confirm the static arrest data (and *vice versa*) and indicate that msR4M-L1 also attenuates MIF-triggered monocyte arrest under flow conditions. Accordingly, these data are also in line with the *ex vivo* arrest data in Fig. 5i-l.

3. The mass spectrum for the main mimic is shown; the mass spectra for the main other mimic as reported on in the table should also be shown (as supplementary data).

Response: In the revised manuscript, we are now including mass spec data as well as the HPLC purification data for msR4M-L2 in new Supplementary Fig. 3.

REVIEWERS' COMMENTS:

Reviewer #1 (Remarks to the Author):

I am impressed and happy with the response of the authors to the issues that were raised by me.

Reviewer #2 (Remarks to the Author):

My comments were addressed satisfactorily.

Reviewer #3 (Remarks to the Author):

The authors have provided new data to address concerns raised in the initial review and removed modeling results which were insufficient to support their structural model. The extensive changes to the figures and comprehensive responses have addressed all my concerns and raised the level of rigor for this study.

Reviewer #4 (Remarks to the Author):

The authors have answered my comments satisfactorily

Response to the Referees' comments

We thank the Referees again for their valuable and constructive comments and their time that have greatly helped us to improve the manuscript.

Reviewer #1:

I am impressed and happy with the response of the authors to the issues that were raised by me.

Response: Thank you very much again for this feedback and for your valuable critique in the initial review.

Reviewer #2:

My comments were addressed satisfactorily.

Response: Thank you very much again for this feedback and for your valuable critique on the initial manuscript version.

Reviewer #3:

The authors have provided new data to address concerns raised in the initial review and removed modeling results which were insufficient to support their structural model. The extensive changes to the figures and comprehensive responses have addressed all my concerns and raised the level of rigor for this study.

Response: Thank you very much again for this feedback and for your valuable critique in the initial review.

Reviewer #4:

The authors have answered my comments satisfactorily.

Response: Thank you very much again for your valuable critique.